# An Actuator Sector Model for Wind Power Applications: A Parametric Study

Mohammad Mehdi Mohammadi[1], Hugo Olivares-Espinosa[1], Gonzalo Pablo Navarro Diaz[1], and Stefan Ivanell[1]

[1]Uppsala University, Department of Earth Sciences

**Correspondence:** Mohammad Mehdi Mohammadi (mohammad.mohammadi@geo.uu.se)

**Abstract.** This paper investigates different actuator sector model implementation alternatives and how they compare to actuator line results. The velocity sampling method, tip/smearing correction, and time step are considered. A good agreement is seen between the line and sector model in the rotor plane and the wake flow. Using the sector model, it was possible to reduce the computational time by 75% compared to the actuator line model as it is possible to run the simulation with a larger time step without compromising the accuracy considerably. The results suggest that the proposed velocity sampling method produces the closest results to the line model with different tip speed ratios. Moreover, the vortex-based smearing correction applied to the sector model results in the lowest error values, among the considered methods, to correct the radial load distributions. Also, it is seen that reducing the time step compared to the one used for the actuator disk/sector does not provide an advantage considering the increased computational time.

## 1 Introduction

The spatial and temporal scales present in simulating the flow field around wind turbines in the atmospheric boundary layer (ABL) can vary from millimeters in the airfoil boundary layer to kilometers when studying wind farms. This puts the direct numerical simulation (DNS) of full Navier-Stokes equations outside of the reach of current high-performance computing (HPC) systems (Sørensen et al., 2015).

In light of this, conducting Large Eddy Simulations (LES) has become a common method to study different areas of interest such as the evaluation of power production and wake interactions under different operational conditions as it can resolve the wake dynamics for farm flows while keeping the computational cost affordable. This is significantly owed to the turbine models representing the complexities of blade-air interactions through simpler means. These models replace the blade geometry with a representative body force distribution in the computational domain. This can be done in a variety of ways.

For instance, in actuator disc models (ADM), the effect of the blades on the incoming wind is represented by body forces distributed asymmetrically over the swept area. After calculating the body forces, they are usually projected into the domain using a distribution such as a 3D Gaussian to avoid numerical instabilities caused by singular momentum sources. An unsteady-3D version of this model is proposed by Sørensen and Myken (1992) showing good agreement with experimental data in

calculating the power values and capturing far wake properties. However, this model is equivalent to an infinitely bladed turbine as it integrates the effect of the blades azimuthally. Therefore, it fails to accurately predict the near wake flow field.

Another alternative is to represent each blade by a line on which the blade forces are calculated. This model was first introduced by Sørensen and Shen (2002) and is referred to as the actuator line model (ALM). ALM is more suitable for studying the near wake properties of the flow in detail and is able to reproduce structures such as tip vortices (Ivanell et al., 2010). However, this model is more computationally demanding partly due to an extra limitation on the simulation time step as a result of the high velocities experienced at the blade tip.

To overcome the additional computational demand of ALM, and the oversimplified representation of blades in ADM, a third alternative has been put forward. Storey et al. (2015) suggests using a sector to represent a blade and showed that this allows marching the solution with a time step similar to an ADM model while solving additional flow structures. The sector is the area swept by a line within the time step used. In this study, this model is referred to as an actuator sector model (ASM). In a study by Nathan et al. (2015), it is shown how the resulted vortex system using this model is similar to that of an ALM. In addition, in separate studies by Krüger et al. (2022) and Vitsas and Meyers (2016), it is shown how ASM can be successfully coupled with an aeroelastic solver.

The mentioned rotor models use a blade-element approach where tabulated airfoil data and local flow field can be used to calculate the radial force distributions along the rotor (Glauert, 1935). It is done by assuming radial and angular elements on the line, sector, or disc. Moreover, a correction for induction effects on the velocities is necessary due to various reasons. For instance, in ADM, a correction is needed to account for a finite number of blades. In comparison, smearing the body forces in ALM calls for another correction to account for the missing induction due to the viscous core behavior of the bound and trailing vortices as explained by (Meyer Forsting et al., 2019). However, in the case of ASM, it is not clear which method would be able to correct the radial distribution of the forces.

Despite the existing studies utilizing an ASM approach, the model implementation itself has not been scrutinized, although it may have significant effects on the results. Therefore, as for the novelty of this work, this paper presents the first comprehensive parametric study on the details of the implementation of an actuator sector model and how they affect the results. This contributes to bridging the current gap in the literature as most studies have left out these details or the choices made are not well justified. It includes the velocity sampling method, tip/smearing correction, rotor updating scheme, and time-step size.

For instance, although Nathan et al. (2015) noticed a significant influence of the velocity sampling method on the torque values and calls for additional investigations, there has not been a study that has addressed this issue to the extent done in this study. Another example is the choice of tip/smearing correction which is usually neglected in the literature. Moreover, a wide range of mesh resolutions are simulated and studied to ensure that the suggested model performs well in all cases and is not specific to one mesh resolution.

An overview of the actuator sector model and the terminology used in this study is provided in section 2. In section 3, the numerical setup and different simulated cases are introduced. The results are presented and discussed in section 4. The findings of the study are summarized and concluded in section 5.

## 2 Model Description

In this section, an overview of the actuator sector model used in this study is presented, and the parameters studied in the parametric study are introduced.

One advantage of using an ASM approach is that the time step for the numerical simulation can be determined from the flow field's CFL condition permitting a larger time step compared to its ALM counterpart. This is achieved by using a sector instead of a line representation of the blades to project the body forces. The time step for ALM and ASM can be obtained from Eqs. (1) and (2), respectively. In Eq. (1), the minimum cell length in the mesh is denoted by $\Delta x$, $\omega$ is the rotational speed, and $R$ is the blade radius. The factor 0.75 is used as a safety factor for ALM while 0.5 is used for ASM to ensure the CFL condition is sustained. The safety factors are determined from preliminary simulations and are needed due to the use of flow properties at the inlet to calculate the time step.

$$\Delta t_{ALM} = 0.75 \cdot \left(\frac{\Delta x}{\omega \cdot R}\right) \tag{1}$$

$$\Delta t_{ASM} = \Delta t_{baseline} = 0.5 \cdot \left(\frac{\Delta x}{U_{incoming}}\right) \tag{2}$$

Compared to the actuator line method, each sector represents a blade and each sector is made of a number of lines starting from line 1 at the beginning of the sector and ending with line $N_{sector}$ at the end of the sector. The sector angle $\theta_{sector}$, the number of lines per sector $N_{sector}$, and the angle between the lines within a sector $\Delta\theta$, are determined using Eqs. (3) to (5). A sector and the representation of its lines and line points are shown in Fig. 1. The sector rotates clockwise with the turbine's rotational velocity and since the sector angle is determined from the time step and $\omega$, the last line in the previous time step is the first line in the current time step.

$$\theta_{sector} = \Delta t \cdot \omega \tag{3}$$

$$N_{sector} = \left(\frac{\theta_{sector} \cdot R}{\Delta x}\right) + 1 \tag{4}$$

$$\Delta\theta = \theta_{sector}/(N_{sector} - 1) \tag{5}$$

Each line is composed of a number of equidistantly distributed points on which the forces are calculated. It is assumed that the contribution of the lines within the sector is equal. This means that the calculated force for each point is divided by the number of lines within the sector. In the first time step, the velocities are read from the computational domain for each line

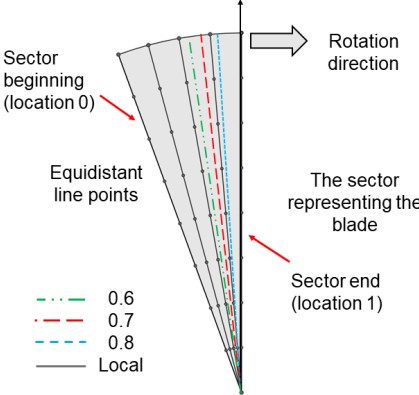

**Figure 1.** A sample sector with 5 lines and 8 line points along with examples of velocity sampling method.

point. This is done by pre-calculating the velocity gradients on the entire disk area. Afterward, the velocity is sampled at the desired coordinate. Then, the forces are calculated based on the angle of attack and the tabulated airfoil data. Afterward, they are projected as body forces using the 3D Gaussian distribution in Eq. (6) where $\epsilon$ controls the smearing length and $r_p$ is the distance from the control point to surrounding cell centers.

In a study by Troldborg (2009), he performed a sensitivity analysis for ALM to determine how the $\epsilon$ value affects the results. He concluded that using $\epsilon = 2\Delta x$ provides an acceptable compromise to reduce the computational time while avoiding numerical instabilities. Moreover, using a fixed $\epsilon$ value for all mesh resolutions would lead to losing some of the flow structures such as tip vortices for finer mesh resolutions due to the over-smearing of the body forces (Martinez et al., 2012; Martínez-Tossas et al., 2015). Therefore, since ASM can be regarded as a sweeping ALM, it is argued that the suggestion by Troldborg (2009) can used to determine the value of $\epsilon$ for ASM in this study.

$$\eta_\epsilon = \frac{1}{\epsilon^3 \cdot \pi^{3/2}} \exp\left(-\frac{r_p^{\,2}}{\epsilon^2}\right) \tag{6}$$

Using ALM, two approaches are conceivable to update the rotor state when the blades rotate. In the first approach, after the body forces are projected into the domain, the velocity field is computed and read from the location of the line points in the current step. Then, the lines are rotated to their new location in the new time step and the blade forces are calculated and projected into the domain. This approach is referred to as the Old Position (OP). In contrast, it is possible to first rotate the lines, then, the velocities are read from the new position, and blade forces are calculated and projected back to the domain. This is referred to as the New Position (NP) approach. It can be argued that these are examples of how the velocity sampling can be done for ALM. However, we use this terminology to be consistent with the code implementation adopted in this study.

The argument in favor of the OP approach is that the self-induction of the blade is zero. In the NP, there is an induced velocity from the blade itself when it is first rotated forward into the next time step. To examine this further, two sets of ALM simulations with different updating schemes and varying $\epsilon$ are conducted. The results are compared with BEM results as it is

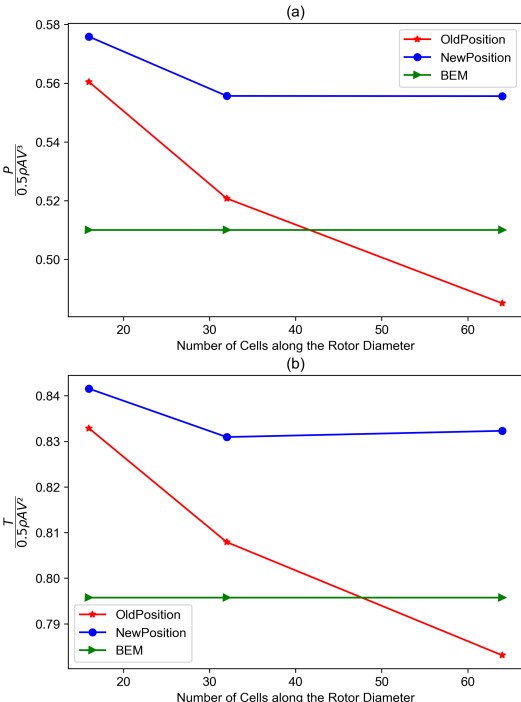

**Figure 2.** The comparison of power and thrust coefficients for OldPostion, NewPosition, and BEM results for different mesh refinements: (a)Power comparison, (b)Thrust comparison

shown previously that it compares well with more sophisticated methods in uniform inflow similar to one considered in the current work (Madsen et al., 2012). The BEM method is implemented according to the algorithm found in (Hansen, 2008). It uses the blade geometry and airfoil properties of the simulated turbine. To account for the finite number of the blades, the Prandtl tip correction is used. However, for induction factors greater than 0.4, Glauert tip loss factor is used instead.

As can be seen in Fig. 2, the power and thrust values have decreased with each refinement for the OP approach, as the forces become more concentrated due to the decreased value of $\epsilon$ which is proportionate to the cell side length. This is on line with the results obtained by Martínez-Tossas et al. (2015). In contrast, using the NP approach, despite reducing the smearing length as the mesh is refined, the values of the thrust and power plateau. The results for both cases are however comparable to BEM results. The choice of updating scheme is of greater importance for a sector compared to a line as its time step and spatial sweep are larger. Moreover, it is not clear to us whether using the NP approach for ASM would produce acceptable results. Therefore, it is decided to present the results using both approaches wherever needed.

Regarding velocity sampling, in ALM, the velocities are usually sampled based on the location of the blade points for each blade. The argument is that using an isotropic Gaussian function as in Eq. (6) to project the body forces results in a circular bound vortex cross-section. Therefore, sampling along the actuator line with the OP approach is equivalent to sampling at the center of the bound vorticity where the blade-local flow effects are not present (Martínez-Tossas et al., 2017). For instance, this

approach has been used in the implementation of ALM in OpenFOAM and EllipSys3D by calculating the velocities at each actuator line point using velocity gradients (Nathan et al., 2017).

Nevertheless, several other methods can be found in the literature in relation to different body force projection functions (Mittal et al., 2015; Churchfield et al., 2017; Xie, 2021). For example, Mittal et al. (2015) suggest using two projection widths based on the local chord and the blade element size along with averaging the velocities from the nearest neighboring cells. Regarding the velocity sampling in ASM, although the body forces for each line within a sector are smeared using an isotropic 3D Gaussian function, the cumulative projection of the body forces does not result in a circular cross-section for the bound

vorticity. Therefore, to find a suitable velocity sampling method that would resemble ALM, several options are tested for each rotor updating scheme.

Three different choices are considered for the NP approach. In the first case, all velocities for all the lines are sampled at the beginning of the sector, i.e. at the first line. The second case samples the velocities for all the lines within the sector from the middle of the sector. If the number of lines is even, the average of velocities read on the two middle lines is used. The third case

uses the local velocity at the location of each line point. These are referred to as 0, 0.5, and local velocity sampling methods, respectively. For the OP approach, 7 cases are considered. It includes sampling at 0.5, 0.6, 0.7, 0.8, 0.9, and 1 of the sector for all lines as well as the local velocity sampling at the location of each line. For instance, 0.5 means the velocities are sampled from the azimuth going through the mid-sector and 1 represents the case where the velocities are sampled from the azimuth at the end of the sector. This is further illustrated in Fig. 1.

Another detail of ASM to investigate is the choice of tip/smearing correction. On one hand, ADM requires a tip correction to take into account the effect of using a finite number of blades. On the other hand, smearing the body forces in the flow domain for ALM to avoid numerical instability results in the overestimation of blade forces near the root and tip of the blades. This is due to the emergence of a viscous core in the released vorticity which in turn reduces the induced velocity at the blade location (Dağ and Sørensen, 2020). Commonly, the corrections proposed by Shen et al. (2005) and Glauert (1935) are used to correct

the calculated forces when using ALM and ADM due to their simplicity and speed (Martinez et al., 2012; Asmuth et al., 2021).

Despite this, they do not produce accurate results as they were originally intended for BEM calculations where the relation between the velocity and forces is not exact. Therefore, in a study by Meyer Forsting et al. (2019), they showed how it is possible to correct the induced velocities for ALM by calculating the missing induction using a near wake model together with a vicious core model. The results showed that the proposed model performs well in a wide variety of operational conditions.

Regarding ASM, there is no consensus on which tip correction would be needed. For this reason, three aforementioned tip corrections are considered to investigate which one would result in satisfactory results for an actuator sector. Although it is conceivable that a tip correction method could be devised for the actuator sector model, this paper aims to determine if currently available methods can correct the load distributions adequately.

It is mentioned earlier that ASM allows for increasing the time step thereby reducing the calculation time. In order to evaluate

the trade-off between the time step size and model accuracy compared to ALM, two smaller time steps than the baseline time step that is calculated from Eq. (2) are considered. This includes $\frac{\Delta t_{baseline}}{2}$ and $\Delta t_{ALM}$.

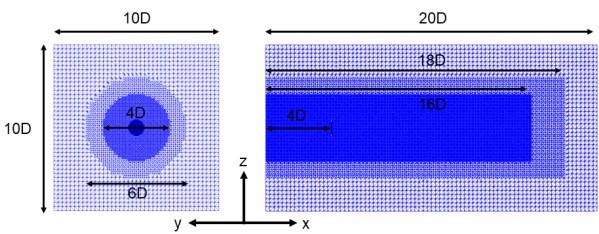

**Figure 3.** The illustration of the computational domain along with the location of the rotor: Left: front-view slice at the rotor plane, Right: side-view at the mid plane

**Table 1.** The sector properties for different mesh resolutions for TSR=7.55

| No. of cells along the rotor diameter | $\theta_{sector}$ | $N_{sector}$ |
|---|---|---|
| 16 | $6.75°$ | 5 |
| 32 | $13.50°$ | 5 |
| 64 | $27.00°$ | 5 |

## 3 Numerical Setup

In this section, the numerical setup used in this study is presented. As explained in section 2, three main parameters in model implementation are investigated. This includes the velocity sampling method, tip/smearing correction, and the time step. More-over, three mesh sizes are run for each case of velocity sampling method to investigate the effect of the mesh size. The details of the simulations used in each section are summarised in Appendix A.

The model is implemented in OpenFOAM by modifying the actuator line implementation found in Simulator fOr Wind Farm Applications (SOWFA) library (Churchfield et al., 2012; Weller et al., 1998). The boundary conditions used are uniform inflow for the inlet, no shear for the outlet, and slip condition for the lateral, upper, and lower sides. The data from the NREL 5MW reference turbine is used for the simulations (Jonkman et al., 2009). The inflow velocity is set to 8 ms$^{-1}$ and the tip speed ratio (TSR) is 7.55.

The used domain is $10D \times 10D \times 20D$ and the turbine is placed 4 diameters downstream of the inlet to minimize the effects from the boundaries. There are two cylindrical refinement areas in the mesh where one is located inside the other one. This way, the cell sides are half and one-fourth of the outermost cell sides for the outer and inner refinement areas, respectively. Therefore, there are 16, 32, and 64 cells per rotor diameter for the coarse to fine mesh, respectively. The computational domain is illustrated in Fig. 3. The sector properties for each of the mesh resolutions used are summarized in Table 1.

The sub-grid scale modeling is done using the modified Smagorinsky model of Mason and Thomson (1992) with model constants of $C_e = 0.93$ and $C_k = 0.0673$. Each case is run for 600 seconds where the results are calculated based on the average of the last 150 seconds corresponding to the time series obtained after flow passing through the entire domain about 3 times. As can be seen in Fig. 4 the power coefficients do not change considerably during this period.

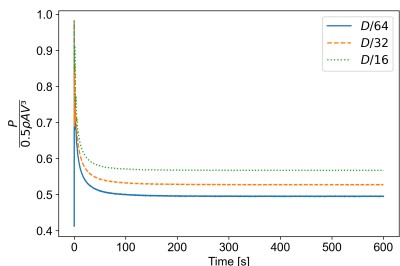

**Figure 4.** The time history of the power coefficient for an ASM case with different mesh resolutions.

Regarding the finite volume schemes, a backward time scheme is used. The interpolation scheme for velocity divergence is local blended linear upwind. This scheme uses 90% linear and 10% upwind for cell face areas equal to or smaller than $(D/32)^2$ while 80% linear and 20% upwind for larger ones. A PGC solver is used for solving the linearized equations with a maximum tolerance of $10^{-6}$. The rest of the parameters are left unchanged as the ones used in the standard ALM settings in SOWFA.

## 4  Result and Discussion

In this section, the results obtained from the parametric study on ASM are presented and discussed. In addition, the results from ADM, ALM, and BEM are provided occasionally to observe how these models compare to the results of ASM. The ALM implementation used in this study is widely utilized in the literature (Asmuth et al., 2021; Martínez-Tossas et al., 2018; Fleming et al., 2015; Churchfield et al., 2012). However, some details can be different due to the different choices available in the turbine model. Moreover, the results of the implementation are already compared with measurements where a good agreement is obtained (Nathan et al., 2017).

First, a comparison of different models regarding the resulting solution in the rotor plane is presented. Second, the results from the investigation of different sampling methods are presented for both OP and NP approaches. This is then supplemented with a sensitivity analysis on the effect of TSR value on the choice of the preferred sampling method. It is followed by evaluating the mentioned tip/smearing corrections. Next, it is tested whether using a smaller time step for ASM compared to its baseline provides an advantage. In the end, the wake profiles and near wake contours of ASM, ALM, and ADM are compared and the computational saving associated with ASM is quantified.

### 4.1  Rotor Plane Solution

Here, the resulting solutions from ASM, ALM, and ADM in the rotor plane are presented. It includes the magnitude of body force, vorticity, and velocity. For conciseness, only the results from the OP approach are presented. The obtained results from the NP approach are similar, although they differ in values. The used mesh has 64 cells along the rotor diameter.

Starting with the body force, as can be seen in Fig. 5, ASM projects a similar distribution as ALM with a distinguished 3-bladed representation of the rotor. However, the forces are more concentrated in ALM as the entirety of the calculated blade

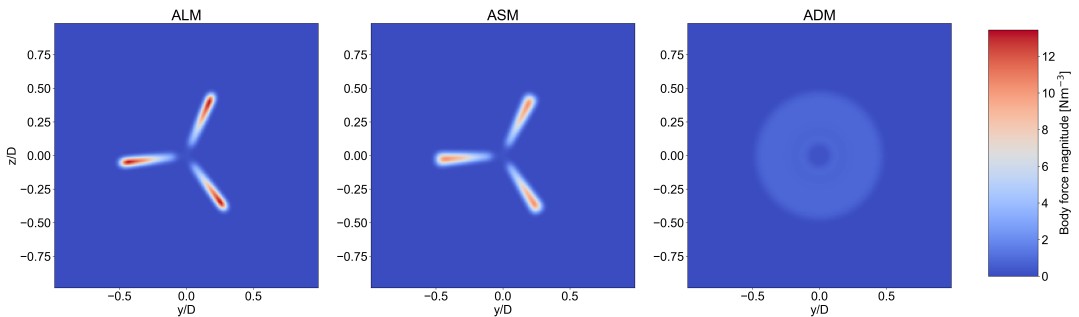

**Figure 5.** Body force distribution in the rotor plane for ALM, ASM, and ADM, respectively from left to right

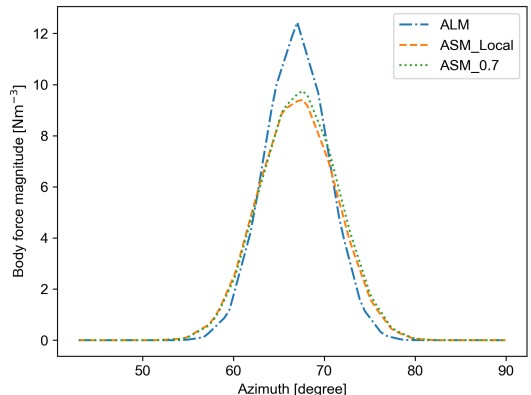

**Figure 6.** Azimuthal distribution of body force for ALM and ASMs with OP approach for the fine mesh with 64 cells along the rotor diameter.

forces is projected from one line while for ASM the forces are split between 5 lines. Looking at the azimuthal distribution of

195 the body forces in the rotor plane as shown in Fig. 6 at 0.8 of the rotor radius, it can be seen how the body force projection of ALM and ASM in the rotor plane differs. ASM has projected the body force in a wider length leading to a lower peak. Nonetheless, the shape of the distributions and the integral of the body forces are comparable.

Regarding the velocity and vorticity magnitude distribution, as seen in Fig. 7 and Fig. 8, ASM has been able to create a similar solution in the rotor plane compared to ALM. The lower extremes seen in ASM compared to ALM are due to the wider

body force projection. Moreover, since the blade forces are projected normal to the rotor plane with the same varying $\epsilon$ which is dependent on $\Delta x$ similar to the one used in ALM, the vortex structures are not diffused in the near wake due to the high values of $\epsilon$ as shown by Martinez et al. (2012). This explains how ASM is able to capture the flow structures in the near wake with an accuracy similar to ALM. These observations provide an indication that using a sector approach, it is possible to recover the ALM solution to a great extent.

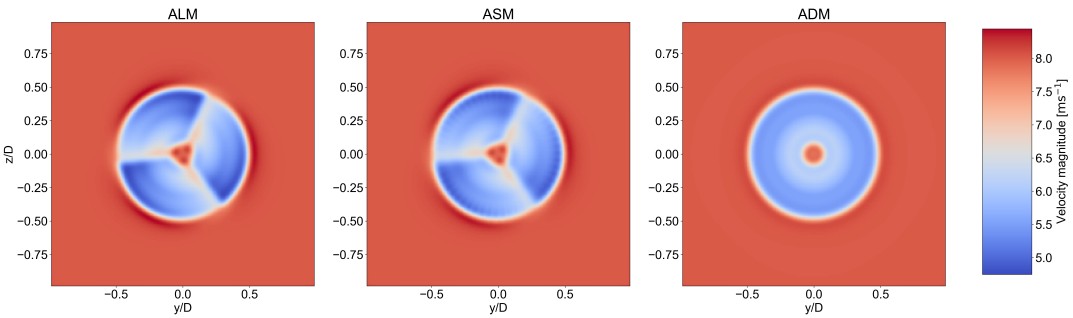

**Figure 7.** Velocity magnitude distribution in the rotor plane for ALM, ASM, and ADM, respectively from left to right

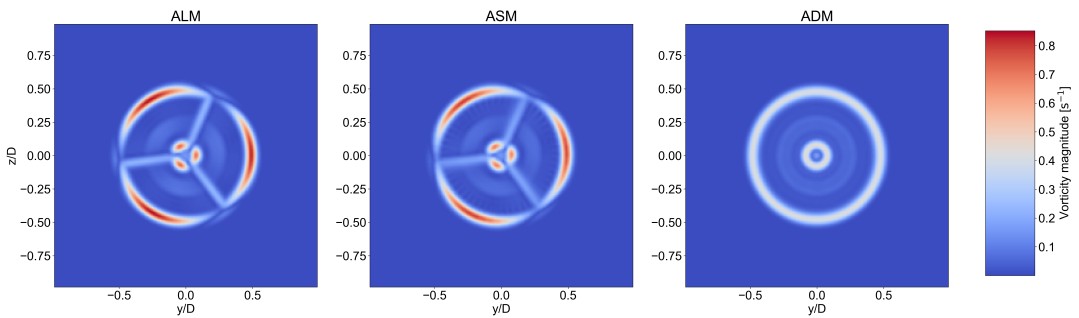

**Figure 8.** Vorticity magnitude distribution in the rotor plane for ALM, ASM, and ADM, respectively from left to right

## 4.2 Velocity Sampling Method

Here, the results from different cases mentioned in section 2 for OP and NP approaches are presented. The cases are run for 3 mesh resolutions with 16, 32, and 64 cells along the diameter, respectively. None of the cases use any tip/smearing correction to isolate only the effect of the velocity sampling method on the results. The ALM results of the same mesh resolution and updating scheme are used for investigating which sampling method matches its ALM counterpart to a greater extent.

### 4.2.1 Old Position Approach

The relative error of power and thrust values is calculated for cases of different mesh resolutions and sampling methods with respect to their ALM counterpart. As can be seen in Fig. 9, the best match with ALM results is achieved for the sampling method where the velocities are sampled at 0.7 of the sector (see Fig. 1) considering all mesh resolutions. For this case, the relative error compared to ALM is around $-1.5\%$ and $-0.5\%$ for power and thrust, respectively. However, it can be argued that for a coarser mesh, other velocity sampling methods could provide closer results to ALM. In that case, Fig. 9 can be used as a guideline.

According to Fig. 10, the power and thrust values have increased with the sampling location moving from 0.5 or mid-section to 1 or the end of the sector. Using the local sampling method has led to values comparable with sampling from the midsection.

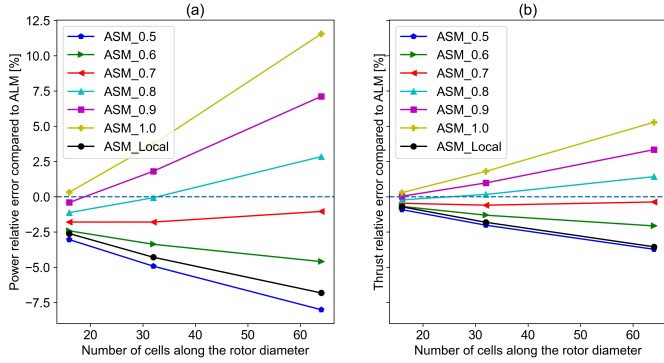

**Figure 9.** Relative errors of power and thrust values of different sampling methods compared to their ALM counterpart of the same mesh resolution with old position updating scheme: (a)Power relative error, (b)Thrust relative error

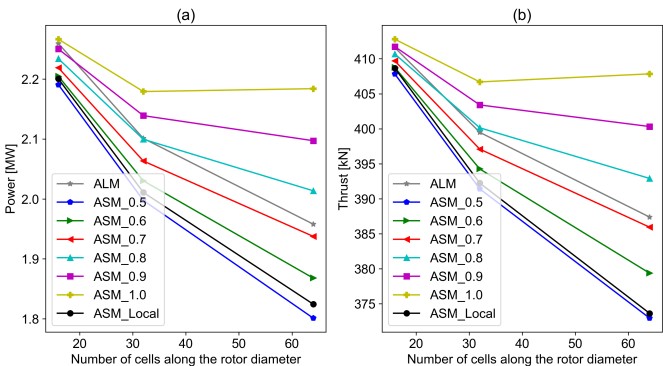

**Figure 10.** Power and Thrust values for ASM of different sampling methods along with their ALM counterpart for different mesh resolutions with old position updating scheme: (a)Power, (b)Thrust

In addition, as shown in Fig. 9 the error values increase sharply with increasing the mesh resolution for other sampling methods.
This shows how relying solely on the results from a relatively coarse mesh can be misleading for tuning this model's parameter.

One of the differences between ALM and ASM is the induction caused on one line by other lines within the sector. However, it is conceivable that there should exist an azimuth within the sector where the cumulative induction from all lines thereby the axial velocity resembles the one obtained by ALM.

Based on our investigation, at 0.7 location within the sector (see Fig. 1), the axial velocity matches best with the one from ALM with OP updating scheme as shown in Fig. 11. Moreover, it is seen how the axial velocity changes throughout the sector on each line. It is noteworthy to mention that although the line forces are equal in each time step for a non-locally sampled model, it does not lead to a symmetric axial velocity azimuthal distribution in the midpoint. This is further illustrated in Fig. 12. In addition, the axial velocity change between one line to the next increases by moving farther from the sector beginning

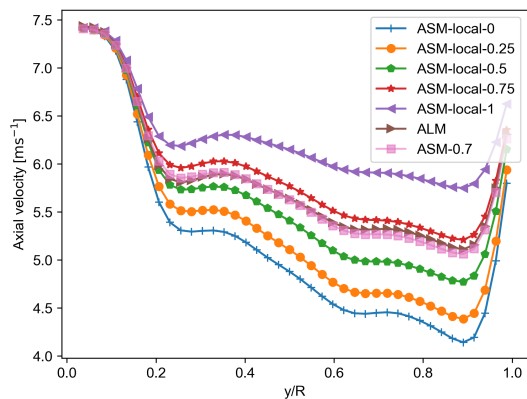

**Figure 11.** Radial distribution of axial velocity on ALM, ASM-local lines, and ASM-0.7 with OP approach for the fine mesh with 64 cells along the rotor diameter.

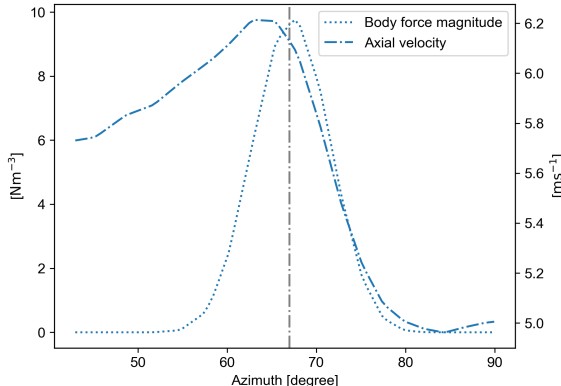

**Figure 12.** Azimuthal distribution of body force and axial velocity for ASM-0.7 with OP approach for the fine mesh with 64 cells along the rotor diameter

thus a higher force and induction. This shows why a locally sampled model such as the one used by Nathan et al. (2015) or from midpoint as Storey et al. (2015) proposed, do not produce the closest results to ALM.

On another note, by recovering the force distribution similar to ALM, by means of sampling from a certain azimuthal position, a similar solution to ALM (in the flow field) is obtained. This could imply that the effect of the different projections of the forces is not as significant compared to the blade force distribution. Therefore, it is conceivable that using an analytical model such as the one described in (Sørensen and Andersen, 2020) would also be practical in ASM when required data for blade element calculations is not unavailable. However, the results will only be as accurate as the model for calculating the blade forces.

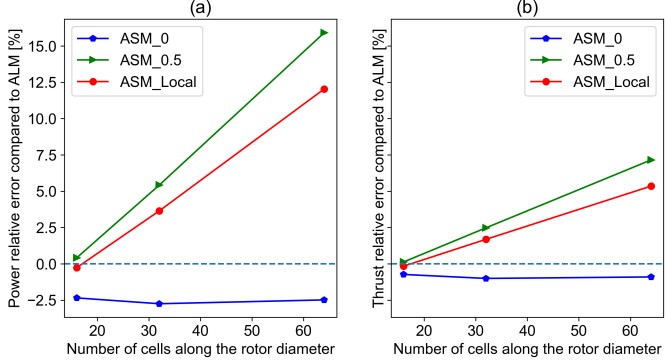

**Figure 13.** Relative errors of power and thrust values of different sampling methods compared to their ALM counterpart of the same mesh resolution with new position updating scheme: (a)Power relative error, (b)Thrust relative error

### 4.2.2 New Position Approach

The power and thrust values for the considered ASM cases with NP updating scheme are presented in Fig. 14 along with their ALM counterpart of the same updating approach for different mesh resolutions. The meaning and the purpose of this updating scheme are explained in section 2. The trend seen in Fig. 2 for ALM with NP updating scheme is only seen for the case where the velocities are sampled at the beginning of the sector. For this case, as can be seen in Fig. 13, the relative error remains below 3% and is around 1% for power and thrust, respectively.

It is noteworthy to mention that this case is equivalent to the OP case where velocities are sampled at the end of the sector. Therefore, one could argue that it is more appropriate to compare the results for this case (NP-0) with ALM with the OP updating scheme. Alternatively, it can be discussed that sampling the velocity at location 1 of a sector with the OP approach produces the closest results to ALM with the NP updating scheme. Similar to 4.2.1, the results of sampling locally or from the mid-section are somewhat alike. For these cases, where the sampling is not done from the sector beginning, the results of ASM with NP approach do not match its ALM counterpart as the velocities are sampled from an azimuth where the blade has not physically reached yet. Moreover, moving the sampling location away from the sector beginning has increased the power and thrust values.

### 4.2.3 Sampling Location Sensitivity to TSR

To ensure that the results obtained from 4.2.1 and 4.2.2 can be generalized to other tip speed ratios relevant to modern wind turbines, additional simulations with two TSR values of 6 and 9 are performed for both ALM and ASM with mesh resolutions of 16, 32, and 64 cells along the rotor diameter while other parameters are kept unchanged. Only the results from the OP updating scheme are presented here. The relative error for power is calculated for the ASM-0.7 case compared to its ALM counterpart of the same TSR and mesh resolution corresponding to the lowest error values for the original TSR of 7.55. As shown in Fig. 15, the error values increase with larger TSR values and vice versa. This is expected as lowering TSR reduces

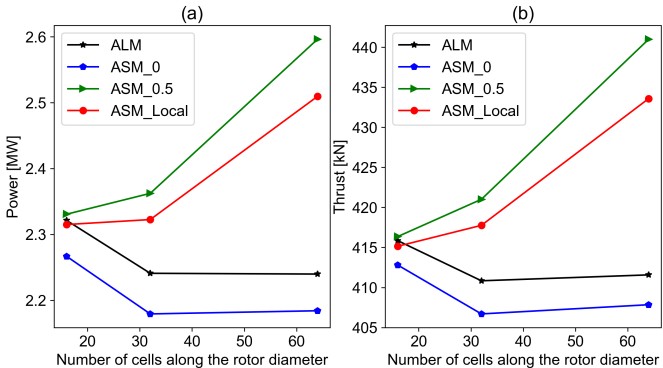

**Figure 14.** Power and Thrust values for ASM of different sampling methods along with their ALM counterpart for different mesh resolutions with new position updating scheme: (a)Power, (b)Thrust

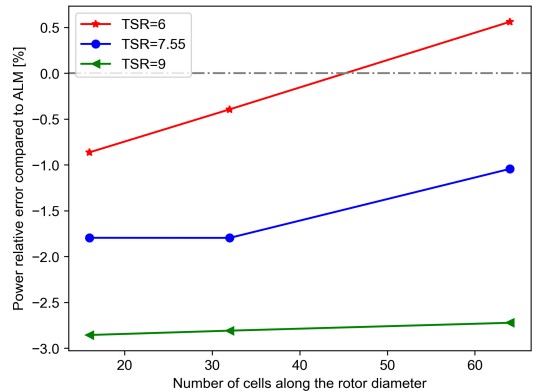

**Figure 15.** Relative error of power value for different TSRs compared to their ALM counterpart of the same mesh resolution

the sector angle, bringing the sector closer to a line. Also, this will further reduce the difference between the time steps used for the line and sector models. Moreover, it is noteworthy to mention that the diverging behavior observed in Fig. 9 for most cases, is not present for the sampling location of 0.7. Therefore, the proposed sampling method is capable of performing well in a range of relevant TSRs.

### 4.3 Tip/Smearing Correction Method

As explained in section 2, three tip/smearing corrections methods have been put to test to investigate which one will suit an ASM model to a greater extent. It includes the corrections by Glauert (1935), Shen et al. (2005), and Meyer Forsting et al. (2020). The sampling method from 0.7 and the beginning (location 0) of the sector are considered for OP and NP schemes, respectively. The investigated mesh has 64 cells along the rotor diameter. The radial distribution of axial and tangential forces are compared to their ALM counterpart of the same mesh resolution and updating scheme. In addition, the load distributions

from BEM method using a Prandtl tip correction are presented to complement the comparisons (Prandtl and Betz, 2010). The used BEM method is introduced in section 2. This will provide an unbiased measure since ALM is also using the vortex-based correction.

Regarding the implementation of these methods, Glauert and Shen corrections use Eq. (7) in which $F_{tip}$ is the loss factor, g is a constant, B is the number of blades, R is the rotor radius, $r_p$ is the blade point radius, and $\phi$ is the angle between the local relative velocity and the rotor plane. There are two main differences between these two corrections. The first one is in the value of constant g. Glauert correction uses the value of 1 while Shen correction determines the value of g using Eq. (8) in which $\lambda$ is the tip speed ratio and $c_1$ and $c_2$ are constants. Using measurement data, Shen et al. (2005) proposed -0.125 and 21 to be the values for $c_1$ and $c_2$ respectively. The value of 0.1 is added to the formulation to prevent it from falling apart for large values of TSR.

$$F_{tip} = \frac{2}{\pi} \times acos(exp(-g \times \frac{B(R-r_p)}{2Rsin\phi})) \tag{7}$$

$$g = exp(c_1 \times (B\lambda - c_2)) + 0.1 \tag{8}$$

The second difference is in the implementation of Glauert correction in SOWFA. It can be argued that since the turbine hub is not modeled, a similar loss factor to Eq. (7) can be calculated for the root section where $(R-r_p)$ is replaced with $(r_p - R_{hub})$ where $R_{hub}$ is the hub radius. Therefore, the total loss factor for Glauert correction is determined using Eq. (9). The calculated loss factors are then multiplied by the drag and lift coefficient obtained from flow information to determine the blade forces. Turning to the vortex-based smearing correction, it is implemented by converting the publicly available original code written in FORTRAN to C++ (Meyer Forsting et al., 2019, 2020).

$$F_{total} = F_{tip} \times F_{root} \tag{9}$$

For the NP scheme, as can be seen in Fig. 16 and Fig. 17, the vortex-based smearing correction has resulted in the closest values to both ALM and BEM outputs with great accuracy. It is not surprising that the correction intended for ALM performs better in comparison to Glauert (1935) or Shen et al. (2005) tip corrections intended for BEM calculations as the sector is more similar to a line than a disc. The mean and maximum relative errors compared to ALM for the vortex-based case are 1.35% and 2.51%, respectively for axial forces. For tangential forces, the mean and maximum relative errors are 2.80% and -5.88%, respectively.

Using Glauert (1935) and Shen et al. (2005) corrections have led to an underprediction of the forces near the blade tip. Moreover, the difference seen in Fig. 17 between Shen and Glauert corrections near the hub can be associated with the explained difference in implementation showing that using an extra loss factor for the blade root might not be necessary. Taking a closer look at the results for the vortex-based correction, it can be seen that although there is an error compared to BEM, the force distribution follows the same trend in the tip region.

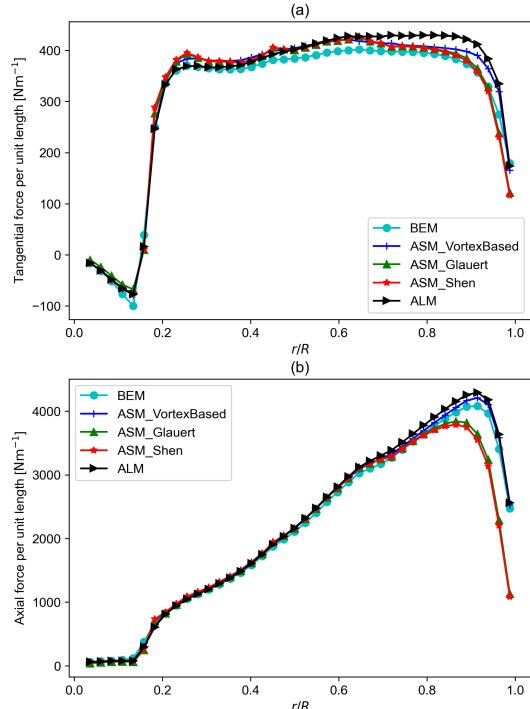

**Figure 16.** Tangential and axial force per unit length comparison with ALM and BEM results with new position updating scheme: (a)Tangential force per unit length, (b)Thrust per unit length

The same trend and findings are true for the OP scheme showing the superiority of the vortex-based smearing correction among the investigated methods and its ability to correct the induced velocities in the rotor plane for an actuator sector. The mean and maximum relative errors compared to ALM for the radial distribution of axial force are 0.57% and 1.17%, respectively. For the tangential force, the mean and maximum relative errors are 1.19% and -2.17%, respectively. The results are presented in Fig. 18 and Fig. 19.

Although the results are satisfactory and comparable to ALM with high accuracy, the larger time step and wider projection of forces in ASM call for an investigation to address and identify the potential effects of these differences to develop a vortex-based smearing correction tailored for ASM applications. Meanwhile, the correction proposed by Meyer Forsting et al. (2020) can be used with enough confidence.

## 4.4 Time-step Size

In this part, the effect of reducing the ASM time step toward its ALM equivalent is investigated to find out if using a smaller time step -thereby a smaller sector angle- can improve the model's accuracy. Although this would remove the model's advantage of reducing the computational power, it provides useful insight into how the model is dependent on the choice of the time step. Two smaller time steps are considered. The smallest time step is equal to $\Delta T_{ALM}$ and the other one is half of the ASM

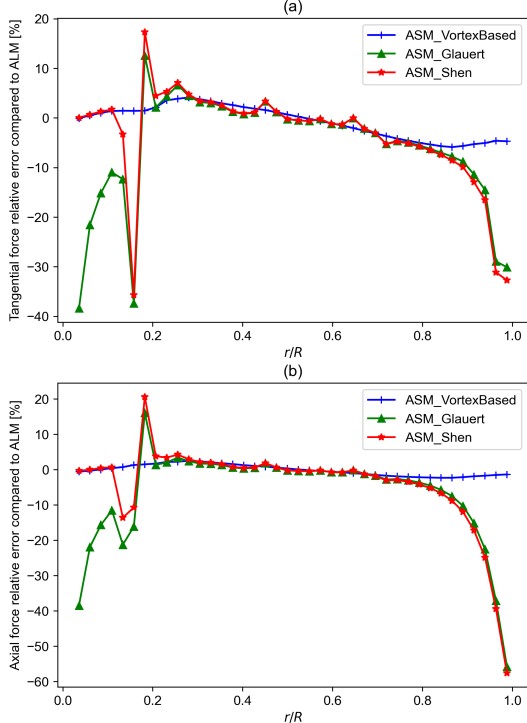

**Figure 17.** Tangential and axial force relative error compared to ALM with new position updating scheme: (a)Tangential force relative error, (b)Axial force relative error

baseline time step. The highest resolution mesh with 64 cells along the rotor diameter is used. For each updating scheme, two sampling methods are considered. It includes 0.7 and local for OP and 0 and local for NP. As a reminder, 0.7 and 0 correspond to sampling the velocity from the 70% and beginning of the sector for all the lines within the sector, respectively and local sampling refers to reading the velocity separately for each line based on their location in the sector.

The power and thrust values for ASMs with different sampling methods and updating schemes are presented in Fig. 20. Moreover, the ALM and BEM results are also provided. As can be seen, reducing the time step and thereby the sector angle has reduced the thrust and power values for the cases with the NP updating scheme. In contrast, the power and thrust values have increased for the cases with the OP updating scheme. For both the OP-0.7 and NP-0 cases that previously showed the best match with their ALM counterparts, the model's accuracy compared to ALM has declined while the accuracy has increased compared to BEM results. Nevertheless, the changes in the values are not significant and are comparable with both ALM and BEM results. For the cases with the local sampling method, there is a great change in the power and thrust values. Reducing the time step has shown a great improvement for both local cases compared to ALM. As can be seen in Fig. 20, running a locally sampled ASM model with the same time step as ALM has almost resulted in the same values since the only difference is that ASM uses two lines compared to ALM with one line.

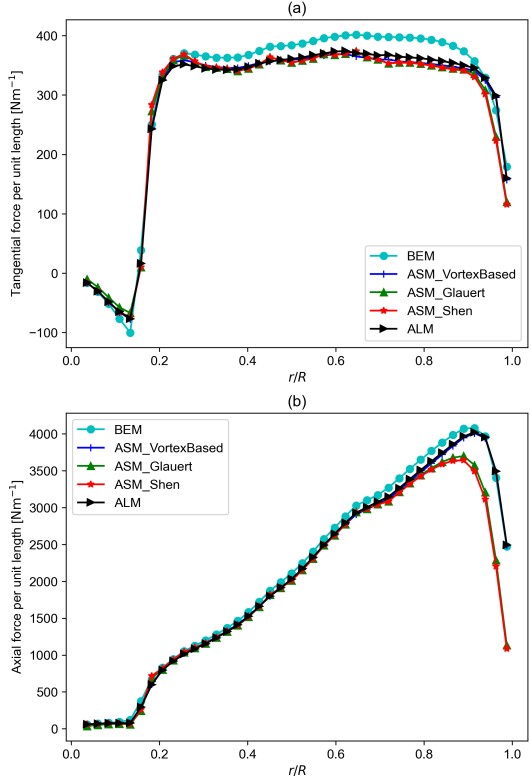

**Figure 18.** Tangential and axial force per unit length comparison with ALM and BEM results with old position updating scheme: (a)Tangential force per unit length, (b)Thrust per unit length

Although reducing the time step takes away the advantage of ASM being faster, it shows how these two models relate to each other. Also, the increase in the error value for the previously tuned sampling location of 0.7 and 0 for OP and NP, can be an indication of how the sampling location is related to the sector angle, number of lines, and their effect on each other through induction. Nonetheless, as the error for thrust and power remained within an acceptable range, the proposed sampling locations may be used in good confidence. Quantifying the dependence of the sampling location on the sector angle shall be investigated in future works.

### 4.5 Near Wake Analysis

In this part, the near wake profile resulting from ASM, ALM, and ADM are presented and compared with each other to investigate to what extent ASM is able to capture the flow structures in the near wake. Also, TKE and stream-wise velocity profiles resulting from each of these models are compared in the streamwise direction. To be concise only the results from the OP approach are presented.

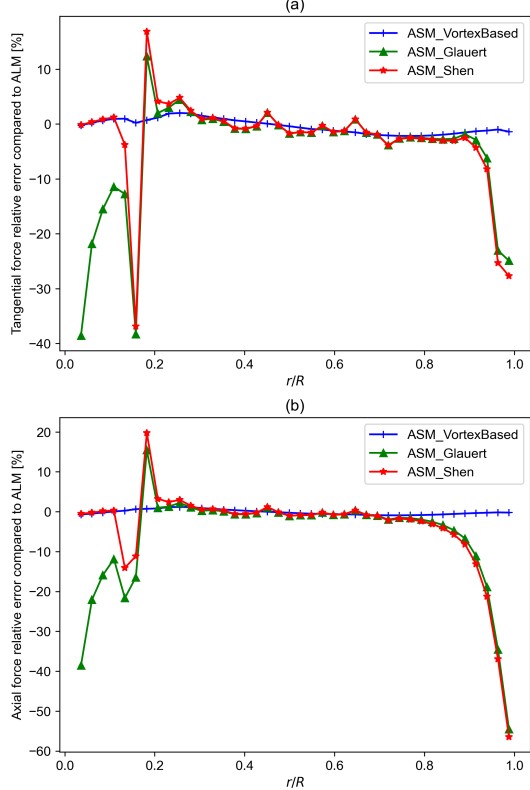

**Figure 19.** Tangential and axial force relative error compared to ALM with old position updating scheme: (a)Tangential force relative error, (b)Axial force relative error

As can be seen in Fig. 21, the helical vortex system captured in ALM is recovered to a great extent by using ASM. In comparison, ADM does not capture any tip vortices and releases a vortex sheet downstream. In spite of capturing the tip vortices in ASM, the maximum value for vorticity is lower compared to ALM as the body forces are less concentrated. The
340 maximum value of vorticity in the near wake of ALM is 0.87 $1/s$ compared to 0.82 and 0.56 for ASM and ADM, respectively.

Looking at the TKE profile of the models for the last 150 seconds of the simulation along the stream-wise direction, as seen in Fig. 22, TKE profiles of ALM and ASM at the rotor agree with each other and are distinct from ADM. However, further downstream, the profiles become similar. For a courser mesh where the tip vortices for ALM and ASM have not formed, TKE values at the rotor are lower than those seen in Fig. 22. Comparing Fig. 22 and Fig. 23, it can be seen that even in the
345 downstream of the rotor, ASM and ALM are more in agreement compared to ADM. However, they become more similar as moving downstream. This is in line with the results obtained by Troldborg et al. (2012). Moreover, using a finer mesh has reduced the difference between the models downstream. The difference in the time-averaged stream-wise velocity profile is however minimal for different models as shown in Fig. 24.

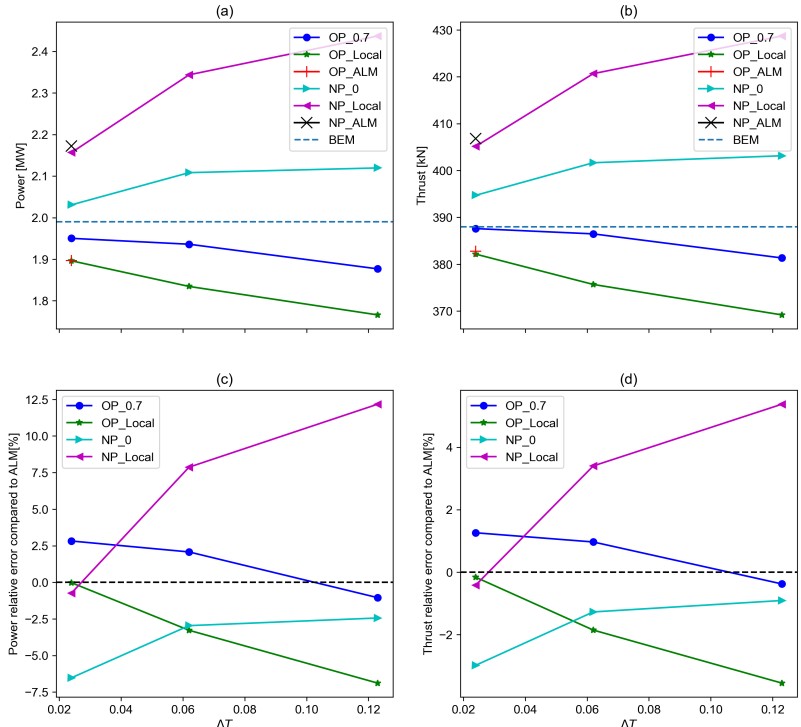

**Figure 20.** The power and thrust values for different ASM/ALM of different sampling methods and updating schemes vs $\Delta T$ along with the power and thrust error values compared to ALM of the same updating scheme vs $\Delta T$: (a)Power vs $\Delta T$, (b)Thrust vs $\Delta T$, (c)Power relative error compared to ALM vs $\Delta T$, (d)Thrust relative error compared to ALM vs $\Delta T$

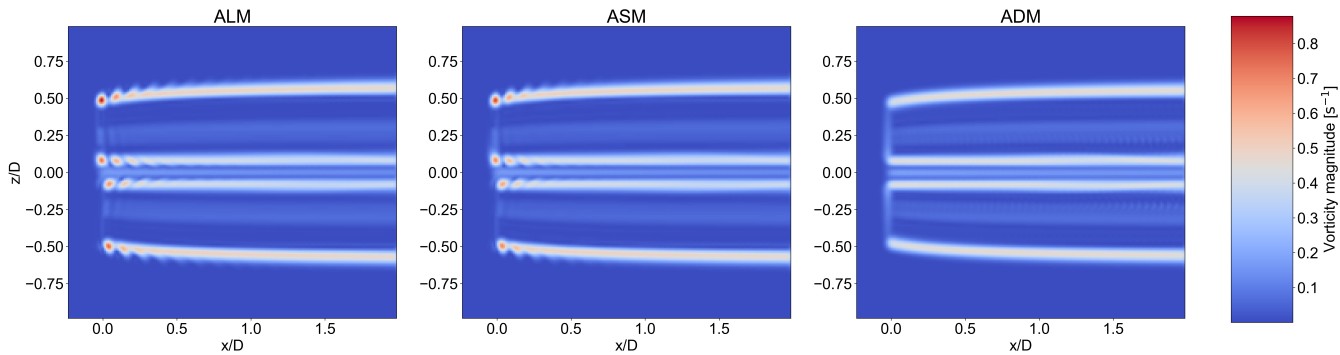

**Figure 21.** Vorticity magnitude along the streamwise direction for ALM, ASM, ADM, respectively from left to right. Fine mesh with 64 cells along the rotor diameter.

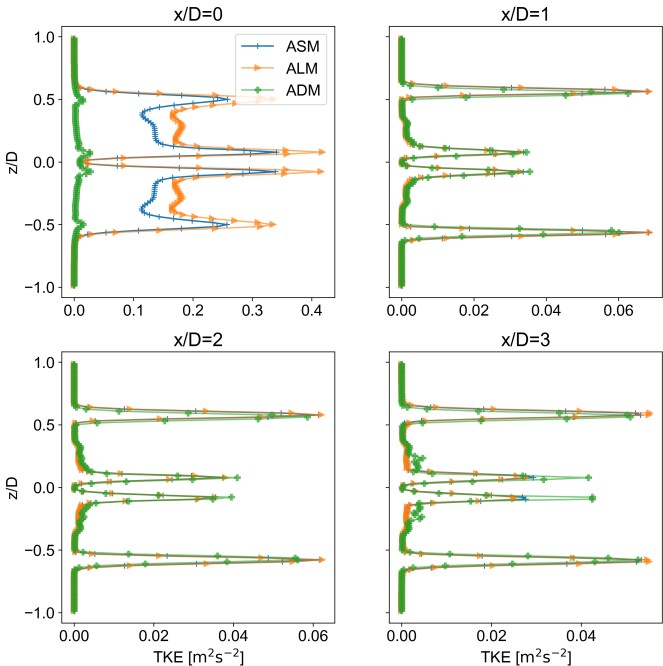

**Figure 22.** TKE profile along the stream-wise direction for ALM, ASM, ADM, fine mesh with 64 cells along the rotor diameter.

## 4.6 Computational Efficiency

Here, a comparison is made between the computational time required for running a similar case with ALM and ASM. Therefore, the mesh and the total simulated time is the same for all cases. Both ALM and ASM models use the vortex-based smearing correction. All models have the OP updating scheme. The ASM model samples the velocities from location 0.7 (see Fig. 1) while sampling for ALM is done locally. The time step used for each of the cases is tabulated in Table 2. Using ASM it has been possible to reduce the computational demand to almost one-fourth or by three-quarters of what is needed for ALM, rendering it to be a faster alternative.

Although the time step used for ASM is 5.125 times larger than the one used for ALM, the CPU time needed for ASM is not as short. This is partly due to the higher number of lines or sub-time steps required in ASM -depending on how the algorithm is implemented- and the associated calculations and tasks. Additionally, this could be due to the shortcomings of the parallelization algorithm used in the flow solver, since the computational time required to solve the flow field is higher compared to the number of calculations needed to compute the body forces (about 1 percent of computations). A further investigation showed that using the sub-time-steps approach can reduce the computational cost almost proportionally to the time-step size.

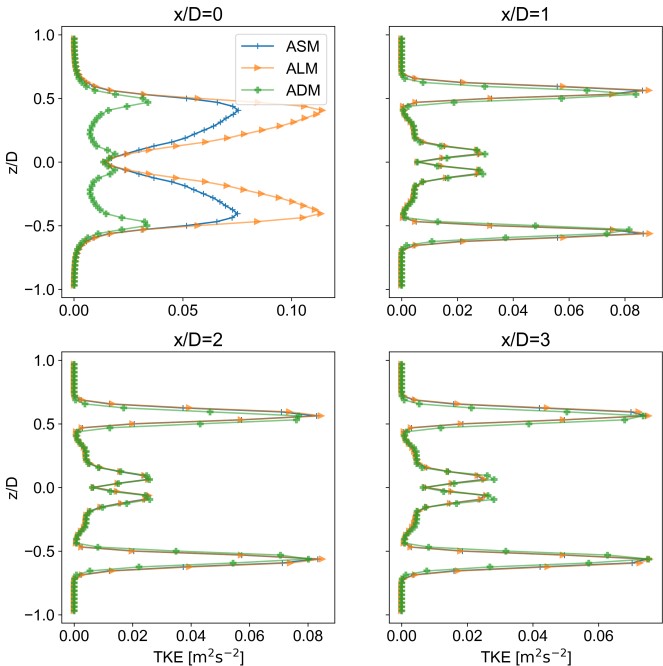

**Figure 23.** TKE profile along the stream-wise direction for ALM, ASM, ADM, moderate mesh with 32 cells along the rotor diameter.

**Table 2.** The time step used for each model

| Model | $\Delta T$ [s] |
|-------|----------------|
| ASM   | 0.123          |
| ALM   | 0.024          |

## 5   Conclusions

The purpose of the current study was to investigate the effects of different available options regarding the actuator sector model
implementation in order to determine the combination leading to the closest results to ALM while improving the computational
speed. It includes the velocity sampling method, tip/smearing correction, and the choice of the time step. To do so, the rotor
plane and near wake solutions resulting from ASM are compared to ALM, ADM, and BEM results where deemed suitable.

Two updating schemes for the rotor state are studied. The updating scheme determines whether the velocities are sampled
before or after the blade rotation to calculate the body forces. Old position (OP) refers to the approach where the velocity
sampling for the current time step is done before rotating the blades into their position in the current time step whereas new
position (NP) refers to sampling velocities for the current time step after the blades are rotated. The study has identified that
sampling the velocities at 70% of the sector angle after the sector beginning with OP updating scheme produces the closest
results to ALM for all considered mesh resolutions. For this case, the relative errors compared to ALM for power and thrust

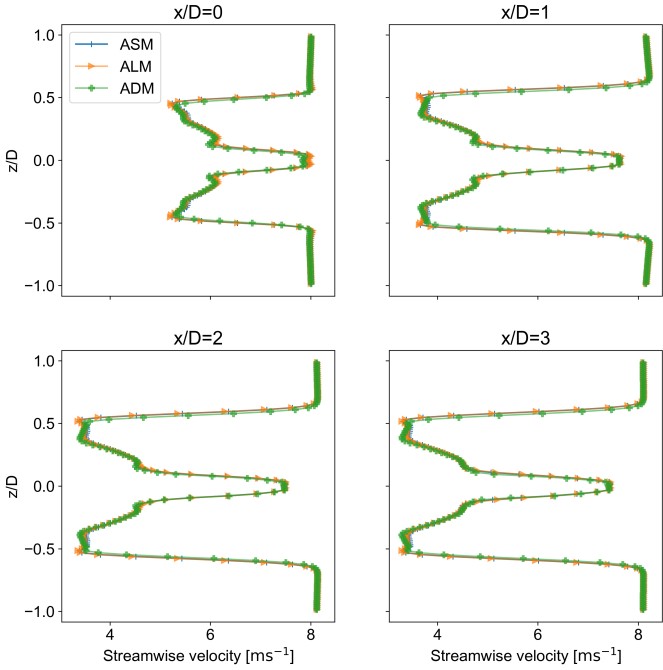

**Figure 24.** Average stream-wise velocity component profile along the stream-wise direction for ALM, ASM, ADM, fine mesh with 64 cells along the rotor diameter.

values are around -1.5% and -0.5%, respectively. For coarser mesh resolutions considered, the results of the study can be used
as a guideline to sample the velocities.

In comparison, when NP updating scheme is used, sampling the velocity from the sector's beginning results in a solution closest to its ALM counterpart. However, as this is equivalent to sampling velocities from the sector's last line with OP approach and a considerable overestimation of the forces was seen for other sampling choices with NP approach, it can be interpreted that sampling the velocity from the last line within the sector with OP updating scheme produces the closest results to an ALM
with NP updating scheme. The relative error values remain below 3% and 1% for power and thrust, respectively for all mesh resolutions. The effect of tip speed ratio (TSR) on the choice of sampling method showed that increasing TSR increases the error values compared to ALM. However, using the recommended approach to sample the velocities keeps the errors within an acceptable range.

The study has confirmed that using the vortex-based smearing correction for ASM will provide the closest load distributions
on the turbine blades compared to its ALM counterpart with great accuracy yielding a mean relative error of 0.57% and 1.2% compared to ALM for radial and tangential load distributions, respectively. This is considerably lower than the other alternatives, especially near the hub and tip regions. Regarding the near wake, TKE profiles of the ASM and ALM showed great agreement with each other and were different than ADM. The difference is at its peak at the rotor plane. Although the difference reduces farther downstream, ASM and ALM were closer to each other. In contrast, the differences in the time-averaged stream-

wise velocity component profiles are minimal between the models. The computational saving of ASM compared to ALM is quantified to be around 75%. Reducing the time step made no significant difference to accuracy considering the reduced speed.

These findings suggest that ASM can potentially replace the commonly used line and disk actuator models for wind power applications as it offers a compromise between computational saving and accuracy compared to ALM as it can capture the near wake dynamics to a better extent than ADM. The present study appears to be the first to investigate the effect of different choices regarding the ASM implementation. This contributes to the adoption of this model by academia as an alternative in the field. However, there are limitations to this study that need to be addressed in future works. The study was limited by the absence of turbulent, yawed, and sheared inflow. Therefore, investigating these cases would be necessary to establish the model's reliability in more realistic conditions. Moreover, although this study used a numerical assessment of different aspects of ASM implementation, it is conceivable to employ an analytical framework for this purpose.

*Code and data availability.* The model implementation code and the simulation data files for different cases resulting from the parametric study can be provided upon request.

## Appendix A: Appendix A

**Table A1.** The details of the simulations used in Fig. 2

| No. | Model | $\Delta x_{min}$ | Tip/Smearing correction | Rotor updating scheme | $\epsilon$ |
|-----|-------|------------------|-------------------------|-----------------------|------------|
| 1 | ALM | $D/16$ | Vortex-based | Old position | $2\Delta x$ |
| 2 | ALM | $D/32$ | Vortex-based | Old position | $2\Delta x$ |
| 3 | ALM | $D/64$ | Vortex-based | Old position | $2\Delta x$ |
| 4 | ALM | $D/16$ | Vortex-based | New position | $2\Delta x$ |
| 5 | ALM | $D/32$ | Vortex-based | New position | $2\Delta x$ |
| 6 | ALM | $D/64$ | Vortex-based | New position | $2\Delta x$ |

**Table A2.** The details of the simulations used in Fig. 4

| No. | Model | $\Delta x_{min}$ | Tip/Smearing correction | Rotor updating scheme | $\epsilon$ | Velocity sampling method |
|-----|-------|------------------|-------------------------|-----------------------|------------|--------------------------|
| 1 | ASM | $D/16$ | Vortex-based | Old position | $2\Delta x$ | 0.7 |
| 2 | ASM | $D/32$ | Vortex-based | Old position | $2\Delta x$ | 0.7 |
| 3 | ASM | $D/64$ | Vortex-based | Old position | $2\Delta x$ | 0.7 |

**Table A3.** The details of the simulations used in Fig. 5,7,8,21,22,24

| No. | Model | $\Delta x_{min}$ | Tip/Smearing correction | Rotor updating scheme | $\epsilon$ | Velocity sampling method |
|-----|-------|------------------|-------------------------|-----------------------|------------|--------------------------|
| 1 | ALM | $D/64$ | Vortex-based | Old position | $2\Delta x$ | Local |
| 2 | ADM | $D/64$ | Shen | Old position | $2\Delta x$ | Local |
| 3 | ASM | $D/64$ | Vortex-based | Old position | $2\Delta x$ | 0.7 |

**Table A4.** The details of the simulations used in Fig. 6

| No. | Model | $\Delta x_{min}$ | Tip/Smearing correction | Rotor updating scheme | $\epsilon$ | Velocity sampling method |
|-----|-------|------------------|-------------------------|-----------------------|------------|--------------------------|
| 1 | ALM | $D/64$ | Vortex-based | Old position | $2\Delta x$ | Local |
| 2 | ASM | $D/64$ | Vortex-based | Old position | $2\Delta x$ | Local |
| 3 | ASM | $D/64$ | Vortex-based | Old position | $2\Delta x$ | 0.7 |

**Table A5.** The details of the simulations used in Fig. 13,14

| No. | Model | $\Delta x_{min}$ | Tip/Smearing correction | Rotor updating scheme | $\epsilon$ | Velocity sampling method |
|-----|-------|------------------|-------------------------|-----------------------|------------|--------------------------|
| 1 | ALM | $D/16$ | none | New position | $2\Delta x$ | Local |
| 2 | ALM | $D/32$ | none | New position | $2\Delta x$ | Local |
| 3 | ALM | $D/64$ | none | New position | $2\Delta x$ | Local |
| 4 | ASM | $D/16$ | none | New position | $2\Delta x$ | 0 |
| 5 | ASM | $D/16$ | none | New position | $2\Delta x$ | 0.5 |
| 6 | ASM | $D/16$ | none | New position | $2\Delta x$ | Local |
| 7 | ASM | $D/32$ | none | New position | $2\Delta x$ | 0 |
| 8 | ASM | $D/32$ | none | New position | $2\Delta x$ | 0.5 |
| 9 | ASM | $D/32$ | none | New position | $2\Delta x$ | Local |
| 10 | ASM | $D/64$ | none | New position | $2\Delta x$ | 0 |
| 11 | ASM | $D/64$ | none | New position | $2\Delta x$ | 0.5 |
| 12 | ASM | $D/64$ | none | New position | $2\Delta x$ | Local |

**Table A6.** The details of the simulations used in Fig. 9,10

| No. | Model | $\Delta x_{min}$ | Tip/Smearing correction | Rotor updating scheme | $\epsilon$ | Velocity sampling method |
|-----|-------|------------------|-------------------------|-----------------------|------------|--------------------------|
| 1 | ALM | $D/16$ | none | Old position | $2\Delta x$ | Local |
| 2 | ALM | $D/32$ | none | Old position | $2\Delta x$ | Local |
| 3 | ALM | $D/64$ | none | Old position | $2\Delta x$ | Local |
| 4 | ASM | $D/16$ | none | Old position | $2\Delta x$ | 0.5 |
| 5 | ASM | $D/16$ | none | Old position | $2\Delta x$ | 0.6 |
| 6 | ASM | $D/16$ | none | Old position | $2\Delta x$ | 0.7 |
| 7 | ASM | $D/16$ | none | Old position | $2\Delta x$ | 0.8 |
| 8 | ASM | $D/16$ | none | Old position | $2\Delta x$ | 0.9 |
| 9 | ASM | $D/16$ | none | Old position | $2\Delta x$ | 1.0 |
| 10 | ASM | $D/16$ | none | Old position | $2\Delta x$ | Local |
| 11 | ASM | $D/32$ | none | Old position | $2\Delta x$ | 0.5 |
| 12 | ASM | $D/32$ | none | Old position | $2\Delta x$ | 0.6 |
| 13 | ASM | $D/32$ | none | Old position | $2\Delta x$ | 0.7 |
| 14 | ASM | $D/32$ | none | Old position | $2\Delta x$ | 0.8 |
| 15 | ASM | $D/32$ | none | Old position | $2\Delta x$ | 0.9 |
| 16 | ASM | $D/32$ | none | Old position | $2\Delta x$ | 1.0 |
| 17 | ASM | $D/32$ | none | Old position | $2\Delta x$ | Local |
| 18 | ASM | $D/64$ | none | Old position | $2\Delta x$ | 0.5 |
| 19 | ASM | $D/64$ | none | Old position | $2\Delta x$ | 0.6 |
| 20 | ASM | $D/64$ | none | Old position | $2\Delta x$ | 0.7 |
| 21 | ASM | $D/64$ | none | Old position | $2\Delta x$ | 0.8 |
| 22 | ASM | $D/64$ | none | Old position | $2\Delta x$ | 0.9 |
| 23 | ASM | $D/64$ | none | Old position | $2\Delta x$ | 1.0 |
| 24 | ASM | $D/64$ | none | Old position | $2\Delta x$ | Local |

**Table A7.** The details of the simulations used in Fig. 11

| No. | Model | $\Delta x_{min}$ | Tip/Smearing correction | Rotor updating scheme | $\epsilon$ | Velocity sampling method |
|-----|-------|------------------|-------------------------|-----------------------|------------|--------------------------|
| 1 | ALM | $D/64$ | none | Old position | $2\Delta x$ | Local |
| 2 | ASM | $D/64$ | none | Old position | $2\Delta x$ | Local |
| 3 | ASM | $D/64$ | none | Old position | $2\Delta x$ | 0.7 |

**Table A8.** The details of the simulations used in Fig. 12

| No. | Model | $\Delta x_{min}$ | Tip/Smearing correction | Rotor updating scheme | $\epsilon$ | Velocity sampling method |
|-----|-------|------------------|-------------------------|-----------------------|------------|--------------------------|
| 1 | ASM | $D/64$ | none | Old position | $2\Delta x$ | 0.7 |

**Table A9.** The details of the simulations used in Fig. 16,17

| No. | Model | $\Delta x_{min}$ | Tip/Smearing correction | Rotor updating scheme | $\epsilon$ | Velocity sampling method |
|-----|-------|------------------|-------------------------|-----------------------|------------|--------------------------|
| 1 | ALM | $D/64$ | Vortex-based | New position | $2\Delta x$ | Local |
| 2 | ASM | $D/64$ | Glauert | New position | $2\Delta x$ | 0 |
| 3 | ALM | $D/64$ | Shen | New position | $2\Delta x$ | 0 |
| 4 | ASM | $D/64$ | Vortex-based | New position | $2\Delta x$ | 0 |

**Table A10.** The details of the simulations used in Fig. 18,19

| No. | Model | $\Delta x_{min}$ | Tip/Smearing correction | Rotor updating scheme | $\epsilon$ | Velocity sampling method |
|-----|-------|------------------|-------------------------|-----------------------|------------|--------------------------|
| 1 | ALM | $D/64$ | Vortex-based | Old position | $2\Delta x$ | Local |
| 2 | ASM | $D/64$ | Glauert | Old position | $2\Delta x$ | 0.7 |
| 3 | ALM | $D/64$ | Shen | Old position | $2\Delta x$ | 0.7 |
| 4 | ASM | $D/64$ | Vortex-based | Old position | $2\Delta x$ | 0.7 |

**Table A11.** The details of the simulations used in Fig. 15

| No. | Model | $\Delta x_{min}$ | Tip/Smearing correction | Rotor updating scheme | $\epsilon$ | Velocity sampling method | TSR |
|---|---|---|---|---|---|---|---|
| 1 | ALM | $D/16$ | none | Old position | $2\Delta x$ | Local | 6 |
| 2 | ALM | $D/16$ | none | Old position | $2\Delta x$ | Local | 7.55 |
| 3 | ALM | $D/16$ | none | Old position | $2\Delta x$ | Local | 9 |
| 4 | ALM | $D/32$ | none | Old position | $2\Delta x$ | Local | 6 |
| 5 | ALM | $D/32$ | none | Old position | $2\Delta x$ | Local | 7.55 |
| 6 | ALM | $D/32$ | none | Old position | $2\Delta x$ | Local | 9 |
| 7 | ALM | $D/64$ | none | Old position | $2\Delta x$ | Local | 6 |
| 8 | ALM | $D/64$ | none | Old position | $2\Delta x$ | Local | 7.55 |
| 9 | ALM | $D/64$ | none | Old position | $2\Delta x$ | Local | 9 |
| 10 | ASM | $D/16$ | none | Old position | $2\Delta x$ | 0.7 | 6 |
| 11 | ASM | $D/16$ | none | Old position | $2\Delta x$ | 0.7 | 7.55 |
| 12 | ASM | $D/16$ | none | Old position | $2\Delta x$ | 0.7 | 9 |
| 13 | ASM | $D/32$ | none | Old position | $2\Delta x$ | 0.7 | 6 |
| 14 | ASM | $D/32$ | none | Old position | $2\Delta x$ | 0.7 | 7.55 |
| 15 | ASM | $D/32$ | none | Old position | $2\Delta x$ | 0.7 | 9 |
| 16 | ASM | $D/64$ | none | Old position | $2\Delta x$ | 0.7 | 6 |
| 17 | ASM | $D/64$ | none | Old position | $2\Delta x$ | 0.7 | 7.55 |
| 18 | ASM | $D/64$ | none | Old position | $2\Delta x$ | 0.7 | 9 |

**Table A12.** The details of the simulations used in Fig. 23

| No. | Model | $\Delta x_{min}$ | Tip/Smearing correction | Rotor updating scheme | $\epsilon$ | Velocity sampling method |
|---|---|---|---|---|---|---|
| 1 | ALM | $D/32$ | Vortex-based | Old position | $2\Delta x$ | Local |
| 2 | ADM | $D/32$ | Shen | Old position | $2\Delta x$ | Local |
| 3 | ASM | $D/32$ | Vortex-based | Old position | $2\Delta x$ | 0.7 |

**Table A13.** The details of the simulations used in Fig. 20

| No. | Model | $\Delta x_{min}$ | Tip/Smearing correction | Rotor updating scheme | $\epsilon$ | Velocity sampling method | $\Delta T$ |
|-----|-------|------------------|-------------------------|-----------------------|-----------|--------------------------|-----------|
| 1 | ALM | $D/64$ | Vortex-based | Old position | $2\Delta x$ | Local | 0.024 |
| 2 | ALM | $D/64$ | Vortex-based | New position | $2\Delta x$ | Local | 0.024 |
| 3 | ASM | $D/64$ | Vortex-based | Old position | $2\Delta x$ | Local | 0.024 |
| 4 | ASM | $D/64$ | Vortex-based | Old position | $2\Delta x$ | 0.7 | 0.024 |
| 5 | ASM | $D/64$ | Vortex-based | New position | $2\Delta x$ | Local | 0.024 |
| 6 | ASM | $D/64$ | Vortex-based | New position | $2\Delta x$ | 0 | 0.024 |
| 7 | ASM | $D/64$ | Vortex-based | Old position | $2\Delta x$ | Local | 0.062 |
| 8 | ASM | $D/64$ | Vortex-based | Old position | $2\Delta x$ | 0.7 | 0.062 |
| 9 | ASM | $D/64$ | Vortex-based | New position | $2\Delta x$ | Local | 0.062 |
| 10 | ASM | $D/64$ | Vortex-based | New position | $2\Delta x$ | 0 | 0.062 |
| 11 | ASM | $D/64$ | Vortex-based | Old position | $2\Delta x$ | Local | 0.123 |
| 12 | ASM | $D/64$ | Vortex-based | Old position | $2\Delta x$ | 0.7 | 0.123 |
| 13 | ASM | $D/64$ | Vortex-based | New position | $2\Delta x$ | Local | 0.123 |
| 14 | ASM | $D/64$ | Vortex-based | New position | $2\Delta x$ | 0 | 0.123 |

*Author contributions.* M. M. Mohammadi and S. Ivanell designed the methodology and M. M. Mohammadi carried them out. M. M. Moham-
madi developed the model code and performed the simulations. G. P. Navarro Diaz contributed to the simulation setup. M. M. Mohammadi
prepared the manuscript. S. Ivanell and H. Olivares-Espinosa provided feedback on it.

*Competing interests.* The authors declare that they have no conflict of interest.

*Acknowledgements.* This work has been funded by Swedish Energy Agency. The simulations were run using resources provided by Swedish
National Infrastructure for Computing (SNIC) and the National Academic Infrastructure for Supercomputing in Sweden (NAISS). Henry
Korb, Henrik Asmuth, and Alexander Meyer Forsting provided feedback on different parts of this study.

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
