# Peer review of "An Actuator Sector Model for Wind Power Applications: A Parametric Study"

_Wind Energy Science, 2023_

## Referee Comment (RC2)

Review of the paper " An Actuator Sector Model for Wind Power Applications: A Parametric Study".

Submitted to **Wind Energy Science**
Article number # : wes-2023-16
Article type: Research article
August 1, 2023

**Recommendation**: Major revisions.

**Summary**:
The manuscript delves into various implementations of actuator sector models and compares them with actuator line results. The primary goal is to assess the accuracy and computational efficiency of the sector model compared to the line model. The paper demonstrates a good agreement between the actuator line and sector models, particularly in the rotor plane and wake flow. Notably, by employing the sector model, the authors achieved a remarkable 75% reduction in computational time compared to the actuator line model. This efficiency gain was made possible by using a larger time step without significantly compromising accuracy. Furthermore, the study highlights that reducing the time step for the actuator disk/sector does not offer a substantial advantage, given the associated increase in computational time. In conclusion, this manuscript provides valuable insights into the implementation and performance of actuator sector models in comparison to actuator line models. The findings shed light on how to achieve efficient yet accurate simulations in wind turbine modeling, which can be of great interest to researchers and engineers in the field of wind energy. Nevertheless, before this manuscript can be considered suitable for publication, several issues need to be resolved. I have divided my comments into two categories: 'Major Concerns' and 'Minor Concerns'. The 'Major Concerns' pertain to conceptual and technical critiques requiring significant attention, while the 'Minor Concerns' draw attention to certain grammatical errors and typos.

**Major concerns**:

- **1**: The manuscript lacks clarity in presenting its novelty. Although it considers three key aspects: velocity sampling method, tip correction, and time step, it is not evident how this work distinguishes itself from other peer papers in the field. The authors should provide a more explicit explanation of the unique contributions of their study.

- **2**: In Figure 3 (The illustration of the computational domain: Left: front-view slice at the rotor plane, Right: side-view at the mid plane), it would be helpful to mark the location/position of the wind turbine in both figures for better clarity and understanding.

- **3**: Finding all the simulations is challenging due to a lack of clear representation. It is suggested to add a table that comprehensively presents the changes made in the simulations, making it easier for readers to understand and follow.

- **4**: The conclusions heavily rely on the comparison between the ASM and ALM models. To strengthen the validity of the ALM simulations, it is important to validate them against other benchmark cases.

- **5**: The section 4.2.2 New Position Approach is confusing and lacks sufficient explanation. The authors should provide further details on this approach to clarify its meaning and purpose.

- **6**: Page 19 mentions the use of load distributions from the BEM method with a Prandtl tip correction for comparisons. It is not clear whether the BEM data was obtained from their paper or calculated independently. If calculated by the authors, additional information is needed to better understand the process.

- **7**: On Page 14, Figure 15, the paper includes two benchmark cases, BEM and ALM. It needs to be specified which one will be used to evaluate the performance of the ALS model.

- **8**: In section 4.3 Tip/Smearing Correction Method, three smearing corrections are used. More detailed information about these three different methods is needed to ensure better clarity and understanding.

- **9**: Figure 16 requires a benchmark case for easy comparison, either BEM or the ALM results should be included.

- **10**: There are no figures for 4.4 Time-step Size. The authors should consider adding appropriate figures to support the discussion in this section.

- **11**: On pages 18-19, Figure 22 shows the TKE profile along the streamwise direction, but it would be beneficial to include a figure showing the streamwise velocity for better context and comparison.

**References**

---

## Author Response (AR1)

**Final response to referee comments**

Mohammad Mehdi Mohammadi[1], Hugo Olivares-Espinosa[1], Gonzalo Pablo Navarro Diaz[1], and
Stefan Ivanell[1]

[1]Uppsala University, Department of Earth Sciences

**Correspondence:** Mohammad Mehdi Mohammadi (mohammad.mohammadi@geo.uu.se)

To begin, I want to thank the referees for their valuable comments on the provided manuscript. We believe that addressing these comments will contribute to the quality of this paper greatly. Moreover, we thank the Copernicus editorial team for their support through this process. In the following, the reviewers' comments are presented followed by our answers in blue. Minimal changes have been applied to the comments to make them compatible with the used text editor. The numbering of figures used in our answers (not the lists) refers to the order in our original submission. Please note that this has changed in the revised version.

Following the guidelines for re-submission of a revised manuscript, we have added a list of relevant changes for each comment after the answer is given. The figures and lines numbering is based on the revised manuscript (without track-change).

**1  RC1**

This manuscript presents a comparison between the actuator line model (ALM) and the actuator sector model (ASM). The article presents interesting insights on the effect of changing the sampling location and time-step in the ASM. There is a fundamental drawback in the method/approach used and the comparisons presented in the manuscript. The authors fix the value of epsilon to epsilon=2dx in their study. This choice limits the validity of the study in 2 ways (https://doi.org/10.1002/we.1747): 1. The results are not expected to converge because epsilon/dx=2 is not enough resolution to resolve the aerodynamics of the blade for that epsilon. The recommended values to be within less than 1% for converged quantities along the blades is epsilon/dx$\geq$5.

Answer: The objective of the study is to suggest an implementation method for ASM that would result in a "converging behavior" when the results are compared to ALM. This means that the relative error for power and thrust values compared to ALM are kept within an acceptable range for different mesh resolutions. The usage of the term "converging" in the manuscript is meant this way. We will emphasize this further in the manuscript. Moreover, the choice of $\epsilon$ is made mainly based on two criteria. The first one is to avoid numerical instability for lower values of $\epsilon$. The second reason is that using a large value of $\epsilon$ such as $\epsilon/dx\geq5$ would result in over-smearing of the body forces in such a way that the tip vortices are not created even for a mesh resolution of 64 cells along the rotor diameter (Martinez et al., 2012; Martínez-Tossas et al., 2015). A choice of $\epsilon/dx=2$ is recommended by Troldborg (2009) based on extensive sensitivity analysis for the value of $\epsilon$ for ALM. This is also pointed out by Martínez-Tossas et al. (2015) that an $\epsilon/dx\geq2$ is required to ensure an oscillation-free solution. Since ASM is basically a

sweeping actuator line or a multi-bladed turbine implemented with ALM, in our judgment, the result of these studies could be applied in our case as well.

The list of relevant changes:

– The reasoning for selecting $\epsilon = 2\Delta x$ is presented in lines 86-91.

– In lines 101-102, it is stated that the reasoning for showing the results in Fig. 2 is to illustrate the applicability of both rotor updating schemes as they compare relatively well with BEM results. It is not our intention to investigate the grid convergence.

– In line 110, the term "converging" is changed to "plateau" to avoid confusion.

– The term "converging behaviour" is changed to "trend" in line 240 to prevent misunderstanding.

– In multiple locations throughout the text, it is mentioned that the results are compared with ALM counterpart of the same mesh resolution and $\epsilon$ value.

2-By fixing epsilon/dx=2, the value of epsilon changes with the grid. Every time the grid is changed, the definition of the problem changes because epsilon is changed. This also leads to differences along the blade of at least a few percent. I recommend the authors to assess these drawbacks and reevaluate the manuscript.

Answer: As explained, the study aims to suggest an implementation for ASM that produces similar results to those obtained by ALM for different mesh resolutions as users may select a different mesh resolution depending on the objective of their study and available computational resources. The motivation is to make sure that the user of the proposed ASM model can be confident that the results obtained by ASM will have an accuracy comparable to ALM for a wide variety of mesh resolutions. Therefore, although the value of $\epsilon$ changes –thereby changing the problem– the results are compared with their ALM counterpart of the same mesh resolution and $\epsilon$, fulfilling the objective of the comparison.

The list of relevant changes:

– All simulations are performed using varying $\epsilon$ value and ASM results are compared to ALM counterpart of the same mesh resolution hence $\epsilon$ value. This is mentioned throughout the text several times. For instance, line 211-212. The details of the simulations can be found in Appendix.

Specific comments: The authors dive into the topic of tip corrections, but this is yet another source of error/difference for the simulations. The results presented have many sources of error and they all contribute to the blade aerodynamics, which makes it too difficult to draw conclusions. The authors are trying to attribute the differences to the choice of sampling location or the width of the ALM, but there are also differences because of the grid resolution and size of epsilon.

Answer: The purpose of the section for tip/smearing correction is to find out, among the considered methods, which one results in the closest results compared to ALM of the same mesh resolution and the same $\epsilon$ value. The smearing correction used for ALM is already validated thoroughly (Meyer Forsting et al., 2019). Therefore, it has been used as a baseline to compare the results of the proposed ASM. In addition, to emphasize further, the results of ASM are compared with ALM of the same mesh

resolution and $\epsilon$ value. Therefore, it is not a major source of difference/error. Moreover, the results of the proposed ASM with the suggested sampling method and smearing correction are the closest to ALM thereby achieving the objective of the study.

60    The list of relevant changes:

– In lines 142-144, the suitability of the vortex-based smearing correction for ALM is pointed out.

– As it has mentioned throughout the text, each ASM case is compared with its ALM counterpart of the same mesh resolution and $\epsilon$ value. For instance, line 267 in the relevant section.

"As can be seen in Fig. 2, the power and thrust values have decreased with each refinement for the OP approach, as the forces become more concentrated due to the decreased value of epsilon which is proportionate to the cell side length" Comment: These figures are showing that there is a grid convergence problem and should not be confused with the old/new velocity sampling. Is epsilon changing as the grid is refined? Can you provide some details of the simulation before showing results?

Answer: The purpose of presenting Fig. 2 is not to evaluate the convergence of ALM but to show that there are two conceivable ways to update the rotor state and depending on which one is selected, the results are different compared to each other. Fig. 2 also uses BEM results to show that although the choices for updating the rotor states are different the results are comparable in both cases with BEM results for different mesh resolutions. Therefore, this motivates why we have considered both of these updating schemes for ASM as well. In the revised manuscript, we have changed the language in a way that this is more clear.

In addition, an extra discussion is made in the revised manuscript regarding the results from ASM with NP approach. It explains that as the only acceptable NP-ASM results are the ones obtained by sampling the velocity from the sector beginning, and this is equivalent to sampling the velocities from the sector's end with OP approach, it can be argued that sampling the velocities from the sector's end produces the closest results to NP-ALM. As mentioned in section 2 (line 81) all simulations in this study are done using $\epsilon=2dx$. This also applies to the simulations used to produce Fig. 2. We will summarize the details of all the conducted simulations in tables as suggested also by Referee 2 in the Appendix.

The list of relevant changes:

80    – As already stated, the purpose of presenting Fig. 2 is not investigating grid convergence but to show how both rotor updating schemes for ALM result in comparable results to BEM. This is explained in lines 100-113.

"Regarding velocity sampling, in ALM, the velocities are sampled on the location of the blade points for each blade. However, in ASM, more methods are conceivable" Comment: There are other methods to sample the velocity in the ALM. Please expand the literature review and cite the work in this area.

85    Answer: This has been a mistake and we will correct this in the revised version of the manuscript. It should have been instead: "Regarding velocity sampling, in ALM, the velocities are usually sampled based on the location of the blade points for each blade" in which "are sampled on" is replaced with "are usually sampled based on". We have also added a few examples of how this is done in the literature in the revised manuscript. The reason why the sampling is done along the actuator line points is that since an isotropic 3D Gaussian function is used to project the forces in the flow domain, the location of actuator line points for the OP approach coincides with the center of the bound vorticity where the flow is not influenced by the blade-local

flow effects for low drag values (Martínez-Tossas et al., 2017). Moreover, an explanation is added as to why this is not the case for ASM since the cumulative body force projection from all lines within a sector does not result in a circular cross-section for the bound vorticity. This also motivates why different velocity sampling methods are tested to investigate which one resembles ALM results to a better extent for different mesh resolutions.

95   The list of relevant changes:

  – The language is changed in line 114 ("usually sampled based on").

  – An explanation is added regarding sampling the velocities along the line in lines 115-119.

  – Additional examples of velocity sampling for ALM are provided in lines 120-122.

"Each case is run for 600 seconds as it is seen the thrust and power values do not change considerably after about 450 seconds

100   which corresponds to flow passing through the entire domain about 3 times." Comment: This is not usual; these simulations typically converge in around 30-60 seconds of simulated time.

Answer: This choice is made to be on the safe side. Otherwise, depending on what threshold is considered for "not changing considerably", even a shorter period of time could be used. Nevertheless, this will not change the results considerably. We will present the time history of the power values for three mesh resolutions to clarify this further. In addition, we have changed the

105   language to "Each case is run for 600 seconds where the results are calculated based on the average of the last 150 seconds corresponding to time series obtained after flow passing through the entire domain about 3 times. As can be seen in Fig. 4, the power coefficients do not change considerably during this period."

The list of relevant changes:

  – The text is changed in lines 168-170.

110   – Fig. 4 is added showing the time history of power coefficient for different mesh resolution.

Comment: Fig 7 is referenced before 5 and 6. Please change the text or the order of the figures.

Answer: We have changed now the order of the figures.

The list of relevant changes:

  – Fig. 6 is now placed between Fig. 5, 7, 8 (rotor plane contours).

115   – It is cross-referenced in line 195 before Fig. 7 and Fig. 8.

"Based on our investigation, at 0.7 the axial velocity matches best with the one from ALM for OP as shown in Fig. 10." Comment: What is 0.7? Please clarify in the text.

Answer: This has been already mentioned in the text a few times for instance for 0.5 and 1 (lines 106 and 107). However, we clarify this further in the text. Moreover, we will show a few of these sampling locations such as 0, 0.5, 0.7, and 1 in Fig.

120   1 which is the sector illustration. For this case, "at 0.7" means the line with an azimuth equal to 70 percent of the sector angle after the sector beginning (location 0).

The list of relevant changes:

125

– Fig. 1 is modified to show several velocity sampling methods.

– Fig. 1 is cross-referenced throughout the text to facilitate the readability of the text. For instance, line 213 and line 224.

125 – Extra information is added in line 224 after 0.7 to clarify this.

"4.2 Velocity Sampling Method" Comment: This section is again mixing the effects of velocity sampling with the vortex-based correction.

Answer: In the beginning, we presented the results for this part without using tip/smearing correction for any of the considered cases. However, we changed them as they are now since we considered that maybe the reader is more interested in seeing

130 the results with the preferred choice of the smearing correction. However, the conclusion drawn from these results does not change. We agree that it could cause confusion for the reader as they have not seen all the results. Therefore, we have revised this part and presented the results from cases without the smearing correction as you mentioned to avoid misunderstanding. Moreover, the figures are replaced accordingly and the reported error values are changed.

The list of relevant changes:

135 – It is added in lines 207-208 that none of the simulations use tip/smearing correction.

– The figures are updated with results of simulations without tip/smearing correction. This includes Figs. 9, 10, 11, 12, 13, 14, and 15.

– In line 214, the relative error for power is updated.

– In line 242, the relative error for thrust is updated.

140 Fig 15 Comment: It is difficult to draw conclusions from these results. All the results differ, but which one is the correct one? The tangential force is different amongst all codes.

Answer: The conclusion drawn in this section for the choice of tip/smearing correction is based on the relative error compared to ALM presented in Fig. 16 and Fig. 18 in the submitted manuscript and Fig. 17 and Fig. 19 in the revised manuscript. It is clear that the relative error for the vortex-based correction has performed well near both hub and tip regions. Using Glauert

145 or Shen correction has led to large error values near both hub and tip regions. As mentioned, the comparison is made relative to an ALM with vortex-based smearing correction as its performance has been validated previously (Meyer Forsting et al., 2019). Moreover, the y-label for Fig. 18 is wrong and it should be tangential relative error instead of axial for (a). This will be corrected in the revised manuscript.

The list of relevant changes:

150 – Fig. 17 and Fig. 19 are used for comparing different methods showing large error values for Glauert and Shen tip corrections.

– Based on Fig. 17 and Fig. 19, the result of ASM with vortex-based smearing correction agree well with its ALM counterpart.

– Based on Fig. 16 and Fig. 18, the vortex-based smearing correction produces the closest results to BEM in the tip region.

"The mean relative errors for radial distribution of axial and tangential forces are 0.57% and 1.20% and the results are presented in Fig. 17 and Fig. 18." Comment: This measurement is misleading, max error would be a better metric. The errors are quite large in some parts of the blade, especially towards the tip.

Answer: The reported error values are for the selected choice of vortex-based smearing correction which performs well all along the blade. We will however add the max error value to better clarify this. The large error values seen near hub and tip regions correspond to the Shen and Glauert tip corrections and this is why they have not been selected in the proposed model in the end.

The list of relevant changes:

– The mean and max relative error values for the vortex-based smearing correction are reported in lines 290-292 and lines 300-301.

**2  RC2**

Summary: The manuscript delves into various implementations of actuator sector models and compares them with actuator line results. The primary goal is to assess the accuracy and computational efficiency of the sector model compared to the line model. The paper demonstrates a good agreement between the actuator line and sector models, particularly in the rotor plane and wake flow. Notably, by employing the sector model, the authors achieved a remarkable 75% reduction in computational time compared to the actuator line model. This efficiency gain was made possible by using a larger time step without significantly compromising accuracy. Furthermore, the study highlights that reducing the time step for the actuator disk/sector does not offer a substantial advantage, given the associated increase in computational time. In conclusion, this manuscript provides valuable insights into the implementation and performance of actuator sector models in comparison to actuator line models. The findings shed light on how to achieve efficient yet accurate simulations in wind turbine modeling, which can be of great interest to researchers and engineers in the field of wind energy. Nevertheless, before this manuscript can be considered suitable for publication, several issues need to be resolved. I have divided my comments into two categories: 'Major Concerns' and 'Minor Concerns'. The 'Major Concerns' pertain to conceptual and technical critiques requiring significant attention, while the 'Minor Concerns' draw attention to certain grammatical errors and typos.

1: The manuscript lacks clarity in presenting its novelty. Although it considers three key aspects: velocity sampling method, tip correction, and time step, it is not evident how this work distinguishes itself from other peer papers in the field. The authors should provide a more explicit explanation of the unique contributions of their study.

Answer: The main novelty of this work is that it presents the first comprehensive parametric study regarding the implementation of the actuator sector model and its effect on the results. Especially regarding the velocity sampling method and tip/smearing correction, there is a knowledge gap in the literature. In most previous works, to the knowledge of the authors, the choice of sampling method is not well justified and even the need for a more detailed study has been pointed out (Nathan

et al., 2015). In addition, other details such as the usage of the tip/smearing correction and rotor updating scheme are left out. Moreover, previous works have often only considered a certain mesh resolution while this study has considered a wide range of mesh resolutions to ensure the suggested velocity sampling method can perform well for all cases. Using the obtained results in this study, it is evident how suggesting a velocity sampling method by using only one mesh resolution can lead to large error values if the mesh resolution is changed. For instance, although sampling the velocities from the mid azimuth line of the sector or locally results in relatively small relative error values for low-resolution mesh cases, the error values increase sharply as the mesh is refined. In summary, the number of implementation details and a wide variety of considered cases in this study provide the potential users with a robust implementation suggestion for the actuator sector model. This motivates those interested to utilize this model for different applications and benefit from its computational saving with greater confidence. We will express the novelty of this work more explicitly in the revised manuscript.

The list of relevant changes:

– Lines 45-54 are added to express the novelty of the work more explicitly.

2: In Figure 3 (The illustration of the computational domain: Left: front-view slice at the rotor plane, Right: side-view at the mid plane), it would be helpful to mark the location/position of the wind turbine in both figures for better clarity and understanding.

Answer: We will add the location of the wind turbine in both figures. Moreover, there is a typo in the front view. The diameters of the inner and outer refinement areas are 4 and 6 diameters, respectively. This will be also corrected in the revised manuscript.

The list of relevant changes:

– Fig. 3 is updated and the location of the turbine is added. The caption is modified accordingly.

– In addition, the diameters shown in the figure are corrected.

3: Finding all the simulations is challenging due to a lack of clear representation. It is suggested to add a table that comprehensively presents the changes made in the simulations, making it easier for readers to understand and follow.

Answer: We will add a table for each section where all simulations are presented along with changes that are made in each of them. This includes the model used (ASM or ALM), velocity sampling method, rotor updating scheme, tip/smearing correction, time step size, tip speed ratio, and mesh resolution. As the tables will occupy too much space for the main text, they will be located in the appendix. The order of the tables is in such a way that minimizes the amount of blank space.

The list of relevant changes:

– The details of all simulations used in each section/ figure are tabulated in Appendix.

– This is mentioned in line 156.

4: The conclusions heavily rely on the comparison between the ASM and ALM models. To strengthen the validity of the ALM simulations, it is important to validate them against other benchmark cases.

Answer: The used ALM implementation and turbine is among the most widely used models in the literature (Asmuth et al., 2021; Martínez-Tossas et al., 2018; Fleming et al., 2015; Churchfield et al., 2012). The code is developed by the National Renewable Energy Laboratory (NREL) and is publicly accessible. In addition, the results from this ALM implementation have been previously compared with measurement and a good agreement was achieved (Nathan et al., 2017). Moreover, using BEM has been shown to produce satisfactory results compared to more sophisticated methods in uniform inflow similar to the case considered in this paper (Madsen et al., 2012). Therefore the results of ALM have been compared to BEM as a reference in Fig. 2 where a good agreement is shown. Hence, we believe that this should be sufficient to use ALM results for comparison, and adding a new section to validate the ALM results is beyond the scope of this work. We will add the reasoning stated here to the revised manuscript to justify why ALM results can be used for comparison. However, we ask the handling associate editor to inform us whether further validation is needed.

The list of relevant changes:

– Lines 102-104 explains why BEM results provided in Fig. 2 and occasionally throughout the paper can be used as an acceptable benchmark.

– Lines 177-181 point out the wide-spread use of the utilized ALM implementation used in this study and how it is previously compared well with measurements.

5: The section 4.2.2 New Position Approach is confusing and lacks sufficient explanation. The authors should provide further details on this approach to clarify its meaning and purpose.

Answer: The updating scheme for the rotor state is explained and the reason why it has been considered in this study is presented in lines 85-100 (section 2: Model description). We will add a cross-reference in the text to enhance readability. We have changed the language in the revised manuscript in such a way that it is more clear. We preferred to not mention this again to avoid repetition. However, if it is needed, we would gladly add a summary at the beginning of this section. Moreover, we have added, that since only the results from ASM with NP approach where the velocities are sampled at the sector beginning are comparable to NP-ALM, and since this is equivalent to sampling the velocities at the sector´s end with OP approach, one could argue that the comparison should be made with OP-ALM. Despite this, we keep the results for the sake of completeness and to adhere to the progression of our investigation, i.e. we did not know from the beginning that this would be the case. The alternative would be to compare the ASM-OP results with both ALM-OP and ALM-NP results. However, we believe that it would be confusing and hard to follow for the reader. In addition, in "Model description" section, we have reformulated the language so it becomes more clear.

The list of relevant changes:

– The language used to explain these two different updating schemes and the motivation for investigating them is reformulated in lines 93-113.

– In lines 239-240, the section where NP approach and its purpose are explained is cross-referenced.

– Lines 243-246 are added to explain how the results from NP approach can be interpreted.

– The interpretation of the results are also added to Conclusions in lines 377-380.

6: Page 19 mentions the use of load distributions from the BEM method with a Prandtl tip correction for comparisons. It is not clear whether the BEM data was obtained from their paper or calculated independently. If calculated by the authors, additional information is needed to better understand the process.

Answer: The BEM data is implemented and calculated by the author following the description in Hansen (2008) using the blade geometry and airfoil properties for NREL5MW turbine. It uses a Prandtl's tip correction to account for the finite number of blades and an empirical correction for the tip loss factor by Glauert for induction factors greater than 0.4. This will be added to the revised manuscript.

The list of relevant changes:

– Lines 104-106 are added to provide information about the implemented BEM code.

7: On Page 14, Figure 15, the paper includes two benchmark cases, BEM and ALM. It needs to be specified which one will be used to evaluate the performance of the ALS model.

Answer: The basis of the evaluation is the comparison with ALM results as shown in Fig. 16 and Fig. 18. The error values reported in the text are based on ALM results as it is mentioned in the text. Moreover, both in the caption and the axis labels for these figures, it is mentioned that the error values are reported compared to ALM.

The list of relevant changes:

– Fig. 17 and Fig. 19 are used for comparing the results. It is mentioned in both captions and y-label that the comparison is made based on ALM results. In addition, it is referred to in the text as well. For instance, line 290, and line 300.

8: In section 4.3 Tip/Smearing Correction Method, three smearing corrections are used. More detailed information about these three different methods is needed to ensure better clarity and understanding.

Answer: We will address this comment by providing a description of the used methods at the beginning of the section. Moreover, reviewing the submitted manuscript, an error was found in the presentation of the results from Shen tip correction. Therefore, the figures and results in this section will be updated in the revised manuscript. The conclusion however will not change as the vortex-based tip correction still shows a better agreement. The description will be similar to the following but more concise.

From a physical perspective, the calculated forces at the blade tip should be zero for a rotor with a finite number of blades as the flow from the pressure and suction sides meet. However, performing BEM calculations without tip corrections results in non-zero values for axial velocity and forces at the blade tip due to the assumption of infinite blades. Both Glauert and Shen tip corrections are intended to account for the finite number of blades in a rotor when performing BEM calculations Shen et al. (2005); Glauert (1935).

To address this, Glauert integrated a correction in BEM calculations to correct the induction velocity in momentum equations while Shen interpreted the correction to be done on airfoil data. The resulting correction is then multiplied by the calculated $C_l$ and $C_d$ values which are the lift and drag coefficients, respectively. This correction should satisfy two criteria. Firstly, it

needs to tend to zero when approaching the tip and secondly, it needs to be 1 for a rotor with an infinite number of blades. The resulting equation for both of these corrections is presented as eq. 1.

$$F_{tip} = \frac{2}{\pi} \times acos(exp(-g \times \frac{B(R-r)}{2Rsin\phi})) \tag{1}$$

where F is the tip loss factor, g is a constant, R is the blade radius, r is the radius at the blade location, and $\phi$ is the angle between the local relative velocity and the rotor plane. The difference between Glauert and Shen corrections is the value of constant g where Glauert used a value of 1 while Shen concluded that this constant is, among others, dependent on the number of blades and tip speed ratio and suggested the eq. 2 for determining its value.

$$g = exp(c_1 \times (B\lambda - c_2)) + 0.1 \tag{2}$$

where $\lambda$ is the tip speed ratio. Using the measured data from two different turbines, values of -0.125 and 21 were found for $c_1$ and $c_2$, respectively. The 0.1 is added to ensure the formulation does not fall apart for extremely large values of tip speed ratio. Another source of difference is in the implementation of SOWFA for Glauert correction. As the turbine hub is not modeled, it can be argued that a correction is needed also for the root of the blades. Therefore, similar to eq. 1, a root loss factor is calculated where $(R-r)$ is replaced with $(r - R_{hub})$ with $R_{hub}$ being the hub radius. In the end, for each blade section, the total loss factor is calculated as eq. 3.

$$F_{total} = F_{tip} \times F_{root} \tag{3}$$

Although both of these corrections are meant for BEM calculations, it is common to use them to correct the forces obtained from Navier-Stokes solvers integrated with ADM and ALM due to its simplicity and low computational cost(Martinez et al., 2012; Asmuth et al., 2021). Therefore, they have been also considered here.

The vortex-based smearing correction considered in this study is however of another nature. Dağ and Sørensen (2020) showed that smearing the blade forces in the CFD domain to avoid numerical instabilities when using ALM results in the presence of a viscous core in the released vorticity. This reduces the induction at the blade location thereby overestimating the angle of attack and the calculated blade forces, especially near the blade root and tip where a large gradient for loads is present. Therefore, Meyer Forsting et al. (2019) presented a correction for ALM where a near wake model is combined with a viscous core model to calculate the missing induction. The results showed that the suggested smearing correction can resolve this issue for a wide range of operational conditions.

In this work, the implementation of this vortex-based smearing correction, publicly available in Meyer Forsting et al. (2019) and originally written in FORTRAN, is translated into C++ and is used along the SOWFA solver for ALM and the developed ASM by the authors to investigate whether it can correct the loading for the proposed ASM.

The list of relevant changes:

- These methods are introduced in lines 135-144.

– Additional information about the methods' implementations are provided in lines 271-285.

315 – Figs. 16, 17, 18, and 19 are updated after solving the issue with the presentation of the Shen tip correction results.

9: Figure 16 requires a benchmark case for easy comparison, either BEM or the ALM results should be included.

Answer: Figure 16 shows the relative error compared to ALM. Therefore, this is already included. Moreover, the results from BEM and ALM are included in Figure 15.

The list of relevant changes:

320 – Fig. 16 and Fig. 18 present the tangential and axial force values including BEM and ALM. Fig. 17 and Fig. 19 present the relative error values compared to ALM.

10: There are no figures for 4.4 Time-step Size. The authors should consider adding appropriate figures to support the discussion in this section.

Answer: Figures 19 and 20 (in the submitted manuscript) are the relevant figures for this section and they are both cited in
325 the text. In the revised manuscript, we put both figures next to each other to increase the readability and save space.

The list of relevant changes:

– In line 316, the relevant figure is cross-referenced.

– The relevant figure for this section is Fig. 20. The previously separate figures for this section are combined and the caption is modified accordingly.

330 11: On pages 18-19, Figure 22 shows the TKE profile along the streamwise direction, but it would be beneficial to include a figure showing the streamwise velocity for better context and comparison.

Answer: Originally, there was also a figure for stream-wise velocity included in this section. However, as the curves were similar, any comparison would be too difficult. Therefore, it was taken away. We will add one figure for the stream-wise velocity in the revised manuscript and will mention the similarity between the different models.
335 The list of relevant changes:

– Fig. 24 is added. It shows the time-averaged stream-wise velocity for different models in the near wake.

– In lines 347-348, the similarity of the profiles are pointed out.

– It is also added to the Conclusion section in lines 389-390.

**3  Final words**

340 We would like to thank the referees and the editorial team again. We did our best to address the received comments. We kindly ask you to inform us about any further required changes and clarifications if necessary.

Kind Regards,

The authors

**References**

Asmuth, H., Navarro Diaz, G. P., Madsen, H. A., Branlard, E., Meyer Forsting, A. R., Nilsson, K., Jonkman, J., and Ivanell, S.: Wind Turbine Response in Waked Inflow: A Modelling Benchmark Against Full-Scale Measurements, SSRN Electronic Journal, https://doi.org/10.2139/ssrn.3940154, 2021.

Churchfield, M., Lee, S., Moriarty, P., Martinez, L., Leonardi, S., Vijayakumar, G., and Brasseur, J.: A Large-Eddy Simulation of Wind-Plant Aerodynamics, in: 50th AIAA Aerospace Sciences Meeting including the New Horizons Forum and Aerospace Exposition, American Institute of Aeronautics and Astronautics, Nashville, Tennessee, https://doi.org/10.2514/6.2012-537, 2012.

Dağ, K. O. and Sørensen, J. N.: A new tip correction for actuator line computations, Wind Energy, 23, 148–160, https://doi.org/10.1002/we.2419, 2020.

Fleming, P., Gebraad, P. M., Lee, S., Van Wingerden, J.-W., Johnson, K., Churchfield, M., Michalakes, J., Spalart, P., and Moriarty, P.: Simulation comparison of wake mitigation control strategies for a two-turbine case: Simulation comparison of wake mitigation control strategies for a two-turbine case, Wind Energy, 18, 2135–2143, https://doi.org/10.1002/we.1810, 2015.

Glauert, H.: Airplane Propellers, pp. 169–360, Springer Berlin Heidelberg, Berlin, Heidelberg, https://doi.org/10.1007/978-3-642-91487-4_3, 1935.

Hansen, M. O. L.: Aerodynamics of wind turbines, Earthscan, London ; Sterling, VA, 2nd ed edn., oCLC: ocm86172940, 2008.

Madsen, H. A., Riziotis, V., Zahle, F., Hansen, M., Snel, H., Grasso, F., Larsen, T., Politis, E., and Rasmussen, F.: Blade element momentum modeling of inflow with shear in comparison with advanced model results: BEM modeling of inflow with shear, Wind Energy, 15, 63–81, https://doi.org/10.1002/we.493, 2012.

Martinez, L., Leonardi, S., Churchfield, M., and Moriarty, P.: A Comparison of Actuator Disk and Actuator Line Wind Turbine Models and Best Practices for Their Use, in: 50th AIAA Aerospace Sciences Meeting including the New Horizons Forum and Aerospace Exposition, American Institute of Aeronautics and Astronautics, Nashville, Tennessee, https://doi.org/10.2514/6.2012-900, 2012.

Martínez-Tossas, L. A., Churchfield, M. J., and Leonardi, S.: Large eddy simulations of the flow past wind turbines: actuator line and disk modeling: LES of the flow past wind turbines: actuator line and disk modeling, Wind Energy, 18, 1047–1060, https://doi.org/10.1002/we.1747, 2015.

Martínez-Tossas, L. A., Churchfield, M. J., Yilmaz, A. E., Sarlak, H., Johnson, P. L., Sørensen, J. N., Meyers, J., and Meneveau, C.: Comparison of four large-eddy simulation research codes and effects of model coefficient and inflow turbulence in actuator-line-based wind turbine modeling, Journal of Renewable and Sustainable Energy, 10, 033 301, https://doi.org/10.1063/1.5004710, 2018.

Martínez-Tossas, L. A., Churchfield, M. J., and Meneveau, C.: Optimal smoothing length scale for actuator line models of wind turbine blades based on Gaussian body force distribution, Wind Energy, 20, 1083–1096, https://doi.org/10.1002/we.2081, 2017.

Meyer Forsting, A. R., Pirrung, G. R., and Ramos-García, N.: A vortex-based tip/smearing correction for the actuator line, Wind Energy Science, 4, 369–383, https://doi.org/10.5194/wes-4-369-2019, 2019.

Nathan, J., Masson, C., Dufresne, L., and Churchfield, M.: Analysis of the sweeped actuator line method, E3S Web of Conferences, 5, 01 001, https://doi.org/10.1051/e3sconf/20150501001, 2015.

Nathan, J., Meyer Forsting, A. R., Troldborg, N., and Masson, C.: Comparison of OpenFOAM and EllipSys3D actuator line methods with (NEW) MEXICO results, Journal of Physics: Conference Series, 854, 012 033, https://doi.org/10.1088/1742-6596/854/1/012033, 2017.

Shen, W. Z., Sørensen, J. N., and Mikkelsen, R.: Tip Loss Correction for Actuator/Navier–Stokes Computations, Journal of Solar Energy Engineering, 127, 209–213, https://doi.org/10.1115/1.1850488, 2005.

Troldborg, N.: Actuator line modeling of wind turbine wakes, Ph.D. thesis, Technical University of Denmark, 2009.

---

## Referee Report (RR1)

Review of the paper "An Actuator Sector Model for Wind Power Applications: A Parametric Study (Revised version 1)".

Submitted to **Wind Energy Science**
October 16, 2023

**Recommendation**: Accept with minor revision.
**Summary**:
I would like to express my gratitude to the authors for their comprehensive response to the comments and for including more detailed explanations in their manuscript. The enhanced discussions, particularly regarding the new Position Approach, have greatly improved the clarity compared to the initial submission. Furthermore, the inclusion of additional details concerning the Tip/Smearing Correction Method has been presented effectively.

In conclusion, I believe the paper is on the right track and can be recommended for publication, pending some minor revisions.

**Minor concerns**:

- **(1)** I appreciate the author's effort to incorporate a table in response to my previous comment #2. Positioning it in the appendix is appropriate. However, I found the format, which lists individual cases for each figure, to be unconventional for a scientific manuscript. I recommend consolidating the information into a single, comprehensive table that encompasses all the cases, rather than presenting them in separate tables.

**References**

---

## Referee Report (RR2)

**Summary:** 'An Actuator Sector Model for Wind Power Applications: A Parametric Study' seeks the optimal settings for minimizing Actuator Sector Model (ASM) error relative to the Actuator Line Model (ALM). Large eddy simulations for both techniques are performed alongside the Actuator Disk Model (ADM) in the Simulator fOr Wind Farm Applications (SOWFA) OpenFOAM library. Results are compared in terms of turbine body forces, power, thrust, wake structure, velocity, vorticity, and turbulence kinetic energy. Optimized ASM outperforms the ADM, especially in the near wake where the ASM recreates the majority of ALM near wake phenomena. With the correct settings, the authors report time saving of up to 75% with low relative errors near 1% across relevant quantities.

**Key Points:** The authors identify the ideal ASM settings to obtain high fidelity LES results for a particular case in substantially less time than with the ALM. This finding is encouraging as balancing computational expense with accuracy is an inherent limitation of all simulations. Overall the manuscript is clear although organization and layout should be improved. Several questions remain with regard to whether the optimal settings are generalizeable and their relationship to underlying ASM theory. The article is primarily concerned with comparing among simulation approaches which limits discussion on the relationship between model parameters and the relevant physics.

**Recommendation:** I recommend publication in *Wind Energy Science* after major revision provided the authors successfully address the comments below.

**Suggestions:**

1. Please consider additional grammar editing.

2. The manuscript needs to be reorganized to improve readability. Figures and tables should be located near where they are first referenced in the text and equations should immediately follow their variable definitions. Searching through the text for variable definitions or the appropriate figure detracts from the quality of the work.

3. "Number of cells along the rotor diameter" is unwieldy as a figure caption and in the text. If the goal is a qualitative comparison between grid resolutions, a naming convention like "Coarse, Medium, Fine" would be more intuitive for the reader. If a quantitative measure is important I, recommend a quantity with physical significance such as $N_{cells}/D$, $\Delta x$, or $\Delta x/D$.

4. The results are well presented and illustrate the impact of various settings on error relative to ALM. However, it is unclear how well they can be generalized to other scenarios without relating optimal parameter settings to underlying physics or ASM theory. For instance, the best sampling location to minimize error appears to be at 0.7 the sector width. If the inflow velocity is increased to 15 m/s, is 0.7 the sector width still the optimal sampling location? If not, is it possible to estimate the correct location from the current results or is a second parameter sweep required?

5. Line 84, please move Equation 6 to the end of this paragraph.

6. Line 100-113 and Figure 2, if these are results please move them to the results portion of the paper.

7. Table 1, why are there 5 sectors for each resolution? Shouldn't the number of sectors increase with $\theta$ following Equation 4? This may also explain why error increases with mesh resolution.

8. Line 166, how many points per line were used for each case? Did the number of points vary with mesh resolution?

9. Lines 271-285 and Equations 7-9, please move this to Section 2. Although it is relevant to the results at hand it interrupts the flow of the manuscript.

---

## Author Response (AR2)

**Response to referee comments: second round**

Mohammad Mehdi Mohammadi[1], Hugo Olivares-Espinosa[1], Gonzalo Pablo Navarro Diaz[1], and Stefan Ivanell[1]

[1]Uppsala University, Department of Earth Sciences

**Correspondence:** Mohammad Mehdi Mohammadi (mohammad.mohammadi@geo.uu.se)

To begin, I want to thank the referees for their valuable comments on the provided manuscript. We believe that addressing these comments will contribute to the quality of this paper greatly. Moreover, we thank the Copernicus editorial team for their support through this process. In the following, the reviewers' comments are presented followed by our answers in blue. Minimal changes have been applied to the comments to make them compatible with the used text editor.

5  Following the guidelines for re-submission of a revised manuscript, we have added a list of relevant changes for each comment after the answer is given. The figures and lines numbering is based on the revised manuscript (without track-change).

**1  RC1**

Minor concerns: (1) I appreciate the author's effort to incorporate a table in response to my previous comment (2). Positioning it in the appendix is appropriate. However, I found the format, which lists individual cases for each figure, to be unconventional

10  for a scientific manuscript. I recommend consolidating the information into a single, comprehensive table that encompasses all the cases, rather than presenting them in separate tables.

Answer: We are glad that you found most of our changes positive and satisfactory. As your suggestion, we change the formatting of the presented tables in Appendix by consolidating them into one table. The column for $\epsilon$ is removed as it is the same for all simulations ($\epsilon = 2\Delta x$). A new column is added for $U_0$ to include cases conducted to respond to comment 4 of the

15  second reviewer. In addition, similar simulations are grouped together to save space.

The list of relevant changes:

- we have changed the format of the tables in the appendix: Simulation Details, by including all cases in one comprehensive table.

**2  RC2**

20  1: Please consider additional grammar editing.

Answer: We do another round of proof reading to improve on this.

The list of relevant changes:

– The revised manuscript is checked for grammatical errors and the corrections are made in the revised manuscript. This can be seen in the marked up version of the manuscript.

25   2: The manuscript needs to be reorganized to improve readability. Figures and tables should be located near where they are first referenced in the text and equations should immediately follow their variable definitions. Searching through the text for variable definitions or the appropriate figure detracts from the quality of the work.

Answer: As your suggestion, we have changed the locations of some figures, tables, and equations to improve readability. We tried to place them as close as possible to where they are mentioned first. However, this was not always possible. In that 30  case, they appear in the relevant section or in the page immediately after. We kindly ask you to inform us if more specific changes are required.

The list of relevant changes:

– Eq. (6) is placed after line 85.

– Fig. 2 orientation is changed to horizontal to save space.

35   – Figs. 5, 6, 7, 8 now appear in the relevant section.

– Figs. 11, 12, 13, 14 appear in the relevant section.

– The layout of Figs. 16, 17, 18, 19 is changed from vertical to horizontal to save space. Now, they appear in the relevant section.

– The layout of Figs. 22, 23, 24 is changed from vertical to horizontal to save space. Figs. 22 and 23 appear in the relevant 40  section and Fig. 24 appears in the page immediately after.

3: "Number of cells along the rotor diameter" is unwieldy as a figure caption and in the text. If the goal is a qualitative comparison between grid resolutions, a naming convention like "Coarse, Medium, Fine" would be more intuitive for the reader. If a quantitative measure is important I, recommend a quantity with physical significance such as $N_{cells}/D$, $\Delta x$, or $\Delta x/D$.

45   Answer: We change the "Number of cells along the rotor diameter" to $D/\Delta x_{min}$. The change is applied throughout the text, figure captions, and figure labels. The coarse-moderate-fine convention is used on occasion as we also think it can be more intuitive.

The list of relevant changes:

– The figure labels have changed for Figs. 2, 9, 10, 13, 14, and 15.

50   – $D/\Delta x_{min}$ is introduced in line 181.

– The header of first column in Table 1 is changed to $D/\Delta x_{min}$.

– The captions of Figs. 6, 11, 12, 21, 22, 23, and 24 are changed accordingly.

– The text is changed accordingly in the first paragraph of 4.1, 4.2, 4.2.3, and 4.3 sections.

4: The results are well presented and illustrate the impact of various settings on error relative to ALM. However, it is unclear how well they can be generalized to other scenarios without relating optimal parameter settings to underlying physics or ASM theory. For instance, the best sampling location to minimize error appears to be at 0.7 the sector width. If the inflow velocity is increased to 15 m/s, is 0.7 the sector width still the optimal sampling location? If not, is it possible to estimate the correct location from the current results or is a second parameter sweep required?

Answer: To address this comment, we conducted two sets of new simulations with ALM and ASM for different mesh resolutions. The first set uses an inflow velocity of 4 m/s and the second case uses an inflow velocity of 20 m/s. The time step size, rotor speed, and pitch angles are changed accordingly. For the 4 m/s case, the TSR value remains constant (TSR=7.55) as it falls into the below-rated operational region. For the 20 m/s case, the TSR is equal to 4. The power relative errors are then computed compared to their ALM counterpart of the same mesh resolution. We added these two new sets to Fig. 15 of the manuscript.

[Figure]

**Figure 1.** Relative error of power value for different TSR and $U_0$ values compared to their ALM counterpart of the same mesh resolution

The results are shown in Fig. 1. As can be seen, for the cases where $U_0$ is constant and only the TSR changes, the error values change. In comparison, for the cases with different $U_0$ and the same TSR, the error values remain the same. This shows that the suitable sampling location depends on the TSR and is not directly affected by $U_0$. Relating this to the ASM theory, it can be shown using ASM equations that changing the TSR value can change the number of lines ($N_{sector}$) within the sector. To do so, we know:

$$\Delta t_{ASM} = \Delta t_{baseline} = 0.5 \cdot (\frac{\Delta x}{U_0}) \tag{1}$$

$$\theta_{sector} = \Delta t \cdot \omega \tag{2}$$

$$N_{sector} = ceil(\frac{\theta_{sector} \cdot R}{\Delta x}) + 1 \tag{3}$$

In addition, from the definition of tip speed ratio (TSR), we have:

$$\omega = \frac{TSR \cdot U_0}{R} \tag{4}$$

75    Substituting eq. 1, eq. 2, and eq. 4 into eq. 3, we get:

$$N_{sector} = ceil(0.5 \cdot TSR) + 1 \tag{5}$$

which shows that the number of lines within the sector only depends on the TSR value. It is expected that changing the number of lines within the sector changes the distribution of the forces thereby changing the induction within the sector. Therefore, the suitable sampling location changes. Despite this, as can be seen in Fig. 1, the error values remain acceptable for
80   a range of relevant TSR values for the suggested sampling method.

[Figure]

**Figure 2.** Relative errors of power and thrust values of different sampling methods compared to their ALM counterpart of the same mesh resolution with old position updating scheme: (a)Power relative error, (b)Thrust relative error

Looking at the error values seen in Fig. 1 (Fig. 15 of the manuscript) and Fig. 2 (Fig. 9 of the manuscript), one could conclude to use the sampling location of 0.6 for $2 < TSR \leq 4$ corresponding to $N_{sector} = 3$. In addition, for $8 < TSR \leq 10$ values yielding $N_{sector} = 6$, the 0.8 sampling location can be adopted to further improve the error values. For most if not all wind turbines, TSR=10 is an upper limit due to various considerations such as noise and safety.

It is perhaps possible to relate the suitable sampling location to the TSR and $N_{sector}$ by fitting a curve to the suitable sampling locations for different $N_{sector}$ or by means of an analytical model. However, this is beyond the scope of this study. We add this as a suggestion for the future work.

In light of this, we rewrite 4.2.3 section and add this discussion. Also, we change the manuscript accordingly where it is needed.

The list of relevant changes:

- The ceiling operator was missing in Eq. (4) of the manuscript and now is added.

- Section 4.2.3 of the manuscript is rewritten.

- Minor changes are applied to section 4.4 (time step size) in light of the new insight.

- Minimal corrections are made to the conclusion section.

- New simulations are added to the Table in Appendix.

5: Line 84, please move Equation 6 to the end of this paragraph.

Answer: we move this equation to the proposed location.

The list of relevant changes:

- It appears now in the suggested location.

6: Line 100-113 and Figure 2, if these are results please move them to the results portion of the paper.

Answer: These are results in the sense that they come from the conducted simulations. However, since they are used to clarify why new and old position updating schemes are considered, we prefer to present them in this section. We think that presenting them in another section could cause confusion. Nevertheless, we kindly ask you to inform us if you think it is required to move them to another section.

7: Table 1, why are there 5 sectors for each resolution? Shouldn't the number of sectors increase with $\theta$ following Equation 4? This may also explain why error increases with mesh resolution.

Answer: As already shown in the response to comment 4 and eq. 5, the number of lines within the sector depends only on the TSR value. Therefore, it does not change with the mesh resolution. In other words, although the $\theta$ changes, since the $\Delta x$ also changes, the number of the lines remains constant.

8: Line 166, how many points per line were used for each case? Did the number of points vary with mesh resolution?

Answer: The same number of points is used for all mesh resolutions. This is equal to 40 points. We add this to the revised manuscript.

The list of relevant changes:

- The number of points is added to line 79.

9: Lines 271-285 and Equations 7-9, please move this to Section 2. Although it is relevant to the results at hand it interrupts the flow of the manuscript.

Answer: We move this part along with the equations to section 2 to avoid interrupting the flow of the manuscript.
The list of relevant changes:

– Lines 271-285 and Equations 7-9 are moved to section 2. Now they appear between line 149-164.

**3   Additional changes**

– The title of appendix is changed to "Simulation Details".

– The variable $U_{incoming}$ is changed to $U_0$. An explanation is added to line 65 to introduce $U_0$ as the inflow velocity.

– The equations use $\cdot$ to show multiplication whereas previously $\times$ was used on occasion.

**4   Final words**

We would like to thank the referees and the editorial team again. We did our best to address the received comments. We kindly ask you to inform us about any further required changes and clarifications if necessary.

Kind Regards,

The authors

---

## Author Response (AR3)

**Response to referee comments: third round**

Mohammad Mehdi Mohammadi[1], Hugo Olivares-Espinosa[1], Gonzalo Pablo Navarro Diaz[1], and Stefan Ivanell[1]

[1]Uppsala University, Department of Earth Sciences

**Correspondence:** Mohammad Mehdi Mohammadi (mohammad.mohammadi@geo.uu.se)

We are pleased that the reviewers found our modifications to the previous manuscript satisfactory. In addition, we would like to thank the referees and the Copernicus editorial team for their valuable time and expertise throughout this process. In the following, the reviewers' minor comments are presented followed by our answers in blue. Minimal changes have been applied to the comments to make them compatible with the used text editor.

Following the guidelines for re-submission of a revised manuscript, we have added a list of relevant changes for each comment after the answer is given. The figures and lines numbering is based on the revised manuscript (without track-change).

**1 RC1**

No further comments were provided.

**2 RC2**

1: Regarding Figure 4, which presents the time history of the power coefficient for an ASM case across various mesh resolutions, it is noted that the text within the figure is too small to be legible.

Answer: The font size is increased to improve readability.

The list of relevant changes:

– The font size in Fig.4 is increased. The new figure replaces the old one.

2: In Figure 4, the power is normalized with $0.5\rho AV^3$, it appears that the variable V has not been explicitly defined.

Answer: We replace the variable $V$ with $U_0$ which is already explicitly introduced in the text as the inflow velocity. In addition, the variable $V$ is replaced with $U_0$ for Fig.2. Moreover, the variables $\rho$ and $A$ are introduced as air density and rotor area respectively in line 107.

The list of relevant changes:

– New Fig.2 and Fig.4 replaced the old ones. Now, variable $V$ is replaced with $U_0$.

– The variables $\rho$ and $A$ are introduced in line 107.

3: It is suggested to include the BEM formulas before using them as a reference, particularly in the context of Figures 16 and 18.

Answer: Following the comment from one of the reviewers in a previous round, we have already added a brief description of the BEM with the relevant citation in the "Model Description" section, lines 104-106. Since the BEM formulas can be readily found in many textbooks and papers, we believe it would be redundant to add them to the text. However, please let us know if you think it is necessary to include them.

**3 Final words**

We would like to thank the referees and the editorial team again. We did our best to address the received comments. We kindly ask you to inform us about any further required changes and clarifications if necessary.

Kind Regards,

The authors